# Glycopeptide epitope facilitates HIV-1 envelope specific humoral immune responses by eliciting T cell help

Lina Sun[1,2], Amy V. Paschall[1,2], Dustin R. Middleton[1,2], Mayumi Ishihara[3], Ahmet Ozdilek[1,2], Paeton L. Wantuch[1,2], Javid Aceil[1,2], Jeremy A. Duke[1,2], Celia C. LaBranche[4], Michael Tiemeyer[1,3] & Fikri Y. Avci[1,2✉]

The inherent molecular complexity of human pathogens requires that mammals evolved an adaptive immune system equipped to handle presentation of non-conventional MHC ligands derived from disease-causing agents, such as HIV-1 envelope (Env) glycoprotein. Here, we report that a CD4[+] T cell repertoire recognizes a glycopeptide epitope on gp120 presented by MHCII pathway. This glycopeptide is strongly immunogenic in eliciting glycan-dependent cellular and humoral immune responses. The glycopeptide specific CD4[+] T cells display a prominent feature of Th2 and Th17 differentiation and exert high efficacy and potency to help Env trimer humoral immune responses. Glycopeptide-induced CD4[+] T cell response prior to Env trimer immunization elicits neutralizing antibody development and production of antibodies facilitating uptake of immunogens by antigen-presenting cells. Our identification of gp120 glycopeptide–induced, T cell–specific immune responses offers a foundation for developing future knowledge-based vaccines that elicit strong and long-lasting protective immune responses against HIV-1 infection.

[1] Department of Biochemistry and Molecular Biology, University of Georgia, Athens, GA 30602, USA. [2] Center for Molecular Medicine, University of Georgia, Athens, GA 30602, USA. [3] Complex Carbohydrate Research Center, University of Georgia, Athens, GA 30602, USA. [4] Department of Surgery, Duke University, Durham, NC 27710, USA. ✉email: avci@uga.edu

Acquired immunodeficiency syndrome (AIDS) caused by human immunodeficiency virus-1 (HIV-1) remains one of the leading causes of death from an infectious agent. Globally, 37.9 million people were living with HIV-1 by the end of 2018 and HIV-1 infection has contributed to more than 35 million deaths since its emergence[1].

The functional envelope spike protein of HIV-1 is a trimer of heterodimers composed of gp120 proteins, each non-covalently associated with a transmembrane gp41 protein and is the primary target for host immune recognition[2–4]. HIV-1 Env trimer is highly glycosylated with glycans contributing to nearly half of its molecular weight[5]. An increasing number of studies have illustrated that in addition to providing a shield to avoid immune responses, gp120 glycans can be the major "sites of vulnerability" targeted by broadly neutralizing antibodies (bNAbs)[6–8], which are the most effective and promising solution for protection against infection and suppression of established HIV-1 infection. In addition, several non-human primate (NHP) studies[9–11] and the RV144 human clinical HIV-1 vaccine trial that exhibited a moderate protective effect against HIV-1 acquisition[12,13] extend our understanding on the correlates of protection against viral challenge conferred by vaccine-induced functional non-neutralizing antibody responses.

Past and current research has so far gained in-depth insight on structural aspects of B-cell receptor/antibody recognition of gp120 epitopes[2,3,6–8]. Major efforts to elicit protective antibody production have been devoted to sophisticated immunogen design[14], especially the development of recombinant native-like Env trimers[15–18]. However, current HIV-1 research does not leverage advances in knowledge related to maximizing stimulation of helper T cells to induce T cell-dependent humoral immune responses to the viral envelope. Despite broad appreciation that glycosylation of HIV-1 gp120 influences the repertoire of antibody responses elicited in infected individuals and that the epitopes recognized by many bNAbs are glycan dependent, the importance of glycopeptides as non-conventional MHC ligands for generating T cell-mediated immunity to HIV-1 has not been addressed[6–8]. To illuminate the mechanisms of T cell-mediated immunity to the HIV-1 envelope, we have explored the interactions of gp120 glycopeptides with the adaptive arm of the immune system. Our previous work has characterized the molecular mechanisms by which glycoconjugate vaccines induce glycan-specific adaptive immune responses[19,20]. Here, we report the existence of a CD4+ T cell repertoire that specifically recognizes gp120 glycopeptide epitopes (i.e., glycotopes). We have identified a gp120 glycopeptide presented by MHCII pathway serving as CD4+ T cell epitope. This glycopeptide elicits a glycan-dependent cellular and humoral immune response. Glycopeptide stimulation also modulates T helper (Th) cell differentiation programming. Functionally, these glycopeptide-specific CD4+ T cells play an important role in helping the humoral immune response to the HIV-1 envelope glycoprotein, which indicates that initiating potent CD4+ T cell responses by glycopeptide epitopes may be an important component of future HIV-1 vaccine strategies.

## Results

**CD4+ T cells recognize glycopeptide-epitopes of gp120.** We first determined whether there exists a population of CD4+ T cells that specifically recognize glycopeptide epitopes on gp120. For generation of immunogens enriched for glycopeptide epitopes, digestion of JR-FL gp120 (clade B) by endoproteinase Glu-C was followed by chromatographic separation and lectin selection (Supplementary Fig. 1a). Immunization of BALB/c mice with pooled gp120 glycopeptides induced gp120-specific IgG production (Supplementary Fig. 1a). CD4+ T cells were isolated from mice immunized with gp120 glycopeptide pool and were stimulated in vitro with intact JR-FL gp120 or with deglycosylated gp120 (DG-gp120)—i.e., gp120 treated with peptide N-glycosidase F (PNGase F) to remove N-glycans (Supplementary Fig. 1b)—in the presence of APCs. Total T cell proliferation was measured as the frequency of carboxyfluorescein diacetate succinimidyl ester (CFSE)low cells. CD4+ T cells stimulated with intact gp120 had a significantly higher proliferation rate, as evidenced by a higher percentage of CFSElow cells in the CD4+ T cell population, than T cells stimulated with DG-gp120 (Fig. 1a, b); this difference provided evidence for a glycopeptide-specific CD4+ T cell proliferative response. To further characterize the gp120 glycopeptide-stimulated T cell population, we collected supernatant from each stimulation group for quantification of interleukin 4 (IL-4) and interferon γ (IFN-γ) production. While CD4+ T cells stimulated with gp120 produced more Th2-associated cytokine IL-4, those stimulated with DG-gp120 produced more Th1-associated IFN-γ (Fig. 1c, d). We next investigated if enriched glycopeptide-specific CD4+ T cell response induces glycan-dependent antibody response against HIV-1 envelope antigen gp120. In a competition ELISA, intact JR-FL gp120 showed much greater inhibition of binding of antisera from mice immunized with the gp120 glycopeptide pool to intact gp120 than DG-gp120 (Fig. 1e). However, the inhibition between intact gp120 and DG-gp120 against sera binding to DG-gp120 was comparable (Supplementary Fig. 1c). These results indicate that substantial glycan-dependent antibodies were elicited after glycopeptide pool immunization. We also assessed the serum IgG subclasses associated with gp120 and DG-gp120 specificity. In line with the cytokine profiles, the gp120-specific IgG response consisted predominantly of the IgG1 subclass, while DG-gp120-binding IgG was predominantly of the IgG2a and IgG3 subclasses (Fig. 1f–h). Taken together, these data demonstrate that certain populations of CD4+ T cells differentiate between glycosylated and non-glycosylated gp120 epitopes, indicating carbohydrate-dependent recognition by CD4+ T cells. Moreover, gp120 glycopeptide stimulation can modulate Th cell differentiation programming.

**Identification of gp120 glycopeptide CD4+ T cell epitopes.** To determine whether gp120 glycotopes are presented by MHCII, we immunoprecipitated MHCII from APCs (mouse bone marrow-derived dendritic cells (BMDCs)) and identified bound epitopes by liquid chromatography coupled to tandem mass spectrometry (LC–MS/MS). BMDCs were co-incubated with glycosylated gp120 expressed in 293-F cells. After cell lysis, MHCII proteins were immunoprecipitated with a monoclonal antibody specific for mouse MHCII molecules I-A and I-E. MHCII-bound epitopes were released and treated with PNGase F in the presence of $^{18}O$-$H_2O$, which marks glycosylated asparagine residues by their conversion to aspartate and incorporation of a heavy oxygen atom. Epitopes released from MHCII were analyzed directly by LC–MS/MS or were digested with one of two proteases (trypsin or chymotrypsin) to enhance sequence coverage and peptide detectability. A variety of glycopeptides and peptides were identified (Supplementary Table 1). One particular glycopeptide epitope derived from the second variable region (V2) of gp120—KLDVVPIDNNN$_{187}$TSYR, glycosylated at N$_{187}$, and designated GpepIP—was identified in both protease-treated and untreated pools of released epitopes, exhibiting single-amino-acid extensions or truncations at the N- or C-terminus (Fig. 2a–c, Supplementary Table 1). This glycopeptide identified from protease-untreated sample represents a naturally processed T cell epitope. As revealed by our previous comprehensive glycomic and glycoproteomic characterization of the gp120 used in this study[21], the predominant glycan structure at the N$_{187}$

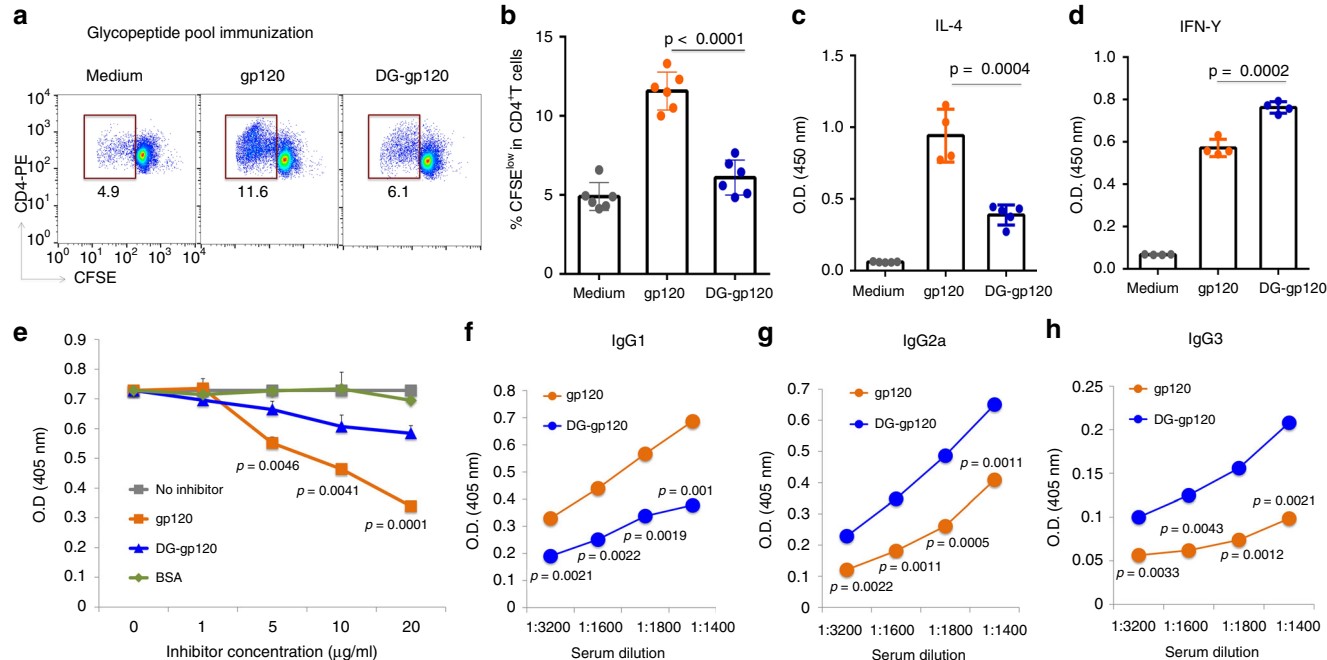

**Fig. 1 gp120 glycopeptide epitopes are recognized by CD4+ T cells.** BALB/c mice were immunized with pooled gp120 glycopeptides (prepared by protease digestion of gp120). After booster immunization, CD4+ T cells were isolated and stimulated in vitro with either intact gp120 or PNGase F–treated, deglycosylated gp120 (DG-gp120) in the presence of mitomycin C-treated APCs for 5 days. **a**, **b** Flow cytometric analysis of T cell proliferation by CFSE division among CD4+ T cells. **c**, **d** Production of IL-4 **c** and IFN-γ **d** in the culture supernatant after T cell stimulation was measured by ELISA. **e** Recognition of coated gp120 by antiserum from mice immunized with gp120 glycopeptides in the presence of the indicated inhibitors was examined by competition ELISA using a serum dilution 1:1600. Serum titers are reported as OD at 405 nm. **f–h** Serum from mice immunized with pooled gp120 glycopeptides were collected 7 days after booster immunization. Titers of IgG1 **f**, IgG2a **g**, and IgG3 **h** for recognition of glycosylated gp120 or deglycosylated gp120 were measured by ELISA. Representative results are shown from one of three independent experiments performed. (mean ± s.d.). **b** n = 6 independent experiments. **c** n = 5 for medium and DG-gp120; n = 4 for gp120. n = 4, 3, 2, 2 independent experiments for **d–h**, respectively. P-values were determined using Student's two-sided t-tests. Source data are provided as a Source Data file.

site is $Man_5GlcNAc_2$ (M5N2, Supplementary Fig. 2) along with other minor oligomannose, hybrid, and complex glycoforms. To obtain the dominant glycoform of this epitope, we recombinantly expressed polyhistidine (His)-tagged GpepIP in N-acetylglucosaminyltransferase I mutant ($GnTI^{-/-}$) cells, which produce exclusively oligomannose N-glycans[22]. In addition, we generated a GpepIP variant with predominantly complex glycans on $N_{187}$ by recombinant expression in 293-F cells. The non-glycosylated peptide backbone (pepIP) was synthesized by solid-state peptide synthesis. All three GpepIP variants were separated by SDS–PAGE, and glycosylation of GpepIP was verified by the molecular-weight shift after PNGase F treatment (Supplementary Fig. 3a, b). GpepIP—but not pepIP—reacted with the mannose-specific lectin concanavalin A (ConA, Supplementary Fig. 3c). The glycan structures on recombinant GpepIP prepared in $GnTI^{-/-}$ cells were confirmed by LC–MS/MS (Supplementary Fig. 4). The major N-glycan on GpepIP was M5N2, with M4N2, M4N2F, and M5N2F as minor components (Supplementary Fig. 4b). GpepIP from 293-F cells expressed almost exclusively complex-type glycans with extensive branching, fucosylation, and sialylation (Supplementary Fig. 5).

To further explore the structural specificity of the $N_{187}$ glycan's impact on GpepIP binding to MHCII, GpepIP variants were exchanged onto recombinant mouse MHCII I-A$^d$ molecules immediately after protease cleavage of CLIP placeholder peptide[23]. MHCII molecules with empty binding grooves aggregate during prolonged incubation[24]; thus, epitope–MHCII binding can be assessed by western blotting for intact MHCII dimer after resolution by isoelectric focusing (IEF) gel electrophoresis (Fig. 2d). OVA peptide 323–339, a well-characterized MHCII-

binding epitope, was used as a positive control, while its scrambled version was used as a negative control for MHCII binding. GpepIP expressed in $GnTI^{-/-}$ cells bound to MHCII with or without a His tag at its N-terminus (Fig. 2d). This observation agrees with the fact that MHCII, structurally different from MHCI, can accommodate larger peptides (i.e., >10 amino acids), as the binding groove is open and allows for greater flexibility[25]. PepIP with a His tag also bound to MHCII as predicted. However, GpepIP expressed in 293-F cells, modified predominantly with complex glycans, did not bind measurably to MHCII. To confirm the presence of MHCII–epitope complexes, we excised bands corresponding to GpepIP/MHCII, pepIP/MHCII, and OVA peptide/MHCII complexes from IEF gels and assayed for peptide and glycopeptide epitopes by LC–MS/MS (Fig. 2e). MHCII α and β chain peptides were also detected (Supplementary Data 1).

**GpepIP elicits glycan-dependent adaptive immune response.** In an assessment of whether GpepIP binding to MHCII elicits cellular and humoral adaptive immune responses, CD4+ T cells isolated from gp120-immunized mice were stimulated with GpepIP or pepIP. The greater T cell proliferation in response to GpepIP than in response to pepIP indicated that GpepIP is a gp120 epitope recognized by CD4+ T cells in a glycan-dependent manner (Fig. 3a, b). CD4+ T cells isolated from mice immunized with $GnTI^{-/-}$-expressed GpepIP were predominantly stimulated by $GnTI^{-/-}$-expressed GpepIP rather than by pepIP (Fig. 3c). This result strongly supports glycan participation in the CD4+ T cell response to the GpepIP epitope; in addition, 293-F-expressed

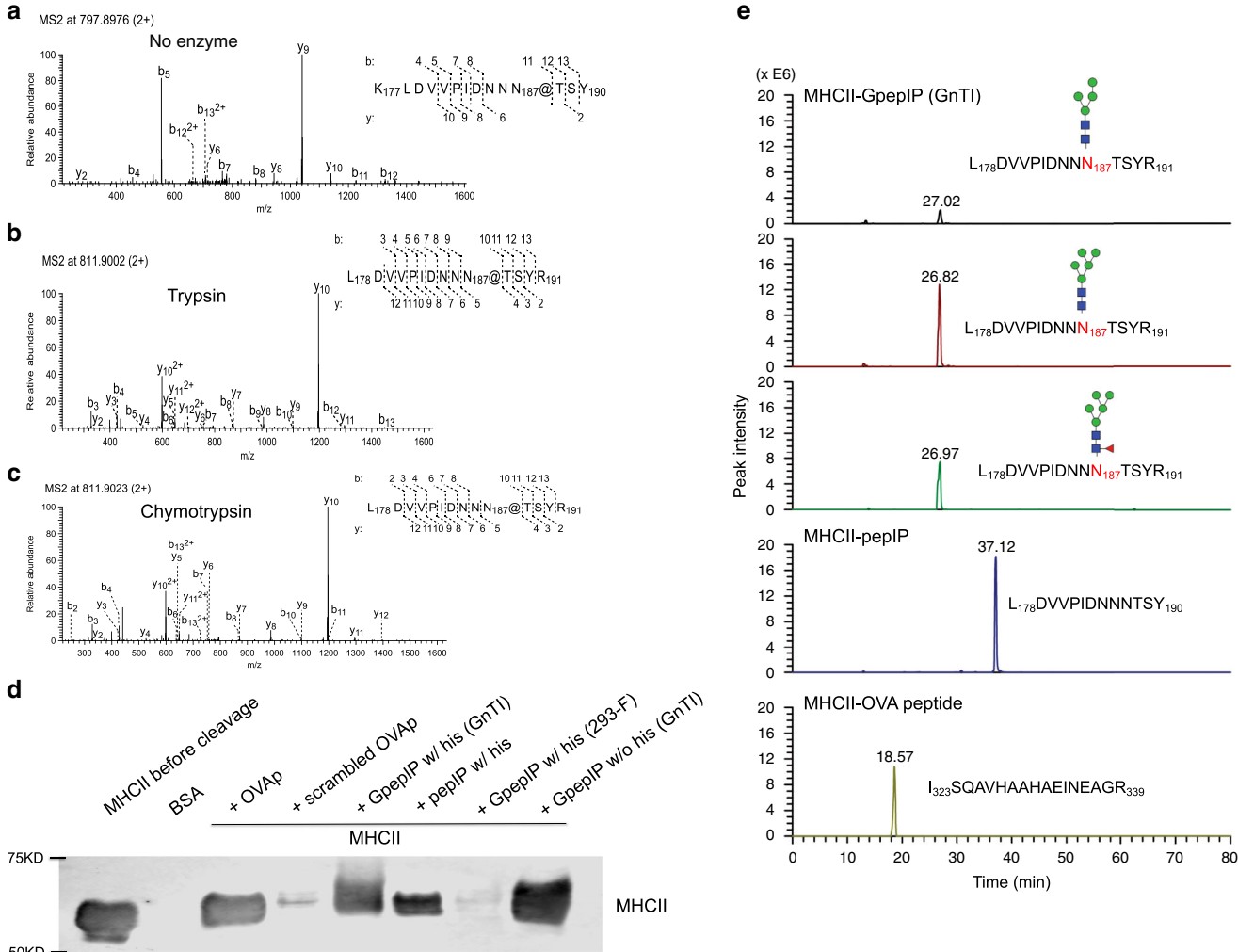

**Fig. 2 Characterization of a gp120 glycotope presented by MHCII. a–c** MS/MS spectra of a gp120 variable-region (V2) glycopeptide (designated GpepIP) identified from MHCII-bound epitopes shows b- and y-ions defining peptide sequence KLDVVPIDNNN$_{187}$TSY in the non-protease group **a**, peptide sequence LDVVPIDNNN$_{187}$TSYR in the trypsin-treated group **b**, peptide sequence LDVVPIDNNN$_{187}$TSYR in the chymotrypsin-treated group **c**, and the N-glycosylation site at N$_{187}$ in all three groups. The @ symbol denotes that the conversion of an Asn residue to Asp with a heavy oxygen was detected at the indicated position. **d, e** Binding of the GpepIP epitope to MHCII molecule was verified. **d** Purified MHCII monomers (mouse allele I-A$^d$) were loaded with an equal amount of the indicated peptides. Peptide-loaded MHCII heterodimers were detected by running of complexes in IEF gel and immunoblotting with mouse MHCII antibody. Representative results are shown from one of three independent experiments performed. **e** Bands corresponding to GpepIP/MHCII, pepIP/MHCII, and OVA peptide/MHCII complexes were excised from IEF gels and subjected to LC-MS/MS analysis. Extracted ion chromatograms demonstrate the binding of three GpepIP glycoforms (M5N2 being the most abundant). PepIP and the positive control (OVA peptide) also bound and stabilized MHCII heterodimers. Source data are provided as a Source Data file.

GpepIP did not stimulate T cell responses (Fig. 3c), possibly because of poor MHCII binding (Fig. 2d). Alternatively, the enhanced potency detected for GnTI$^{-/-}$-expressed GpepIP over that of 293-F cells can be due to T cell receptor recognition of specific oligomannosidic glycan motifs. When mice were immunized with pepIP, their CD4$^+$ T cells responded only to peptide without glycans; adding glycans of any type to this epitope (GpepIP expressed in GnTI$^{-/-}$ or 293-F cells) blocked pepIP-specific CD4$^+$ T cell responses (Fig. 3d). We next explored the requirement for MHCII during CD4$^+$ T cells response to GpepIP and pepIP. CD4$^+$ T cells from GpepIP or pepIP immunized mice were stimulated with GpepIP or pepIP, respectively, in the presence of anti-MHCII antibody. The results clearly showed that MHCII blocking significantly inhibited CD4$^+$ T cells response in a dose-dependent manner (Supplementary Fig. 6a, b). These results revealed that recognition of the GpepIP epitope by CD4$^+$ T cells is both glycan and MHCII dependent.

We next evaluated T cell-dependent antibody responses induced by GpepIP and pepIP immunization in terms of binding to glycosylated and non-glycosylated epitopes. Antisera from GpepIP-immunized mice exhibited remarkably strong binding to GpepIP (Fig. 3e), whereas pepIP immunization predominantly produced high-affinity antibodies to pepIP (Fig. 3f). Importantly, antisera from GpepIP-immunized mice showed significantly greater binding to intact gp120 expressed in either 293-F or GnTI$^{-/-}$ cells than pepIP antisera (Fig. 3g), suggesting that GpepIP is more immunogenic than pepIP in eliciting gp120-specific antibody response. Lack of a differential ELISA response for GnTI$^{-/-}$ gp120 and even for 293-F gp120 is likely due to oligo/high-mannose structures displayed on gp120, including the GpepIP site (Supplementary Fig. 2). We also assessed the IgG subclasses of GpepIP-immunized antisera associated with gp120 and DG-gp120 specificity. GpepIP immunization induced higher IgG1 and IgG2a subclasses of gp120-specific IgG than DG-gp120-specific IgG subclass (Fig. 3h, i). gp120-specific

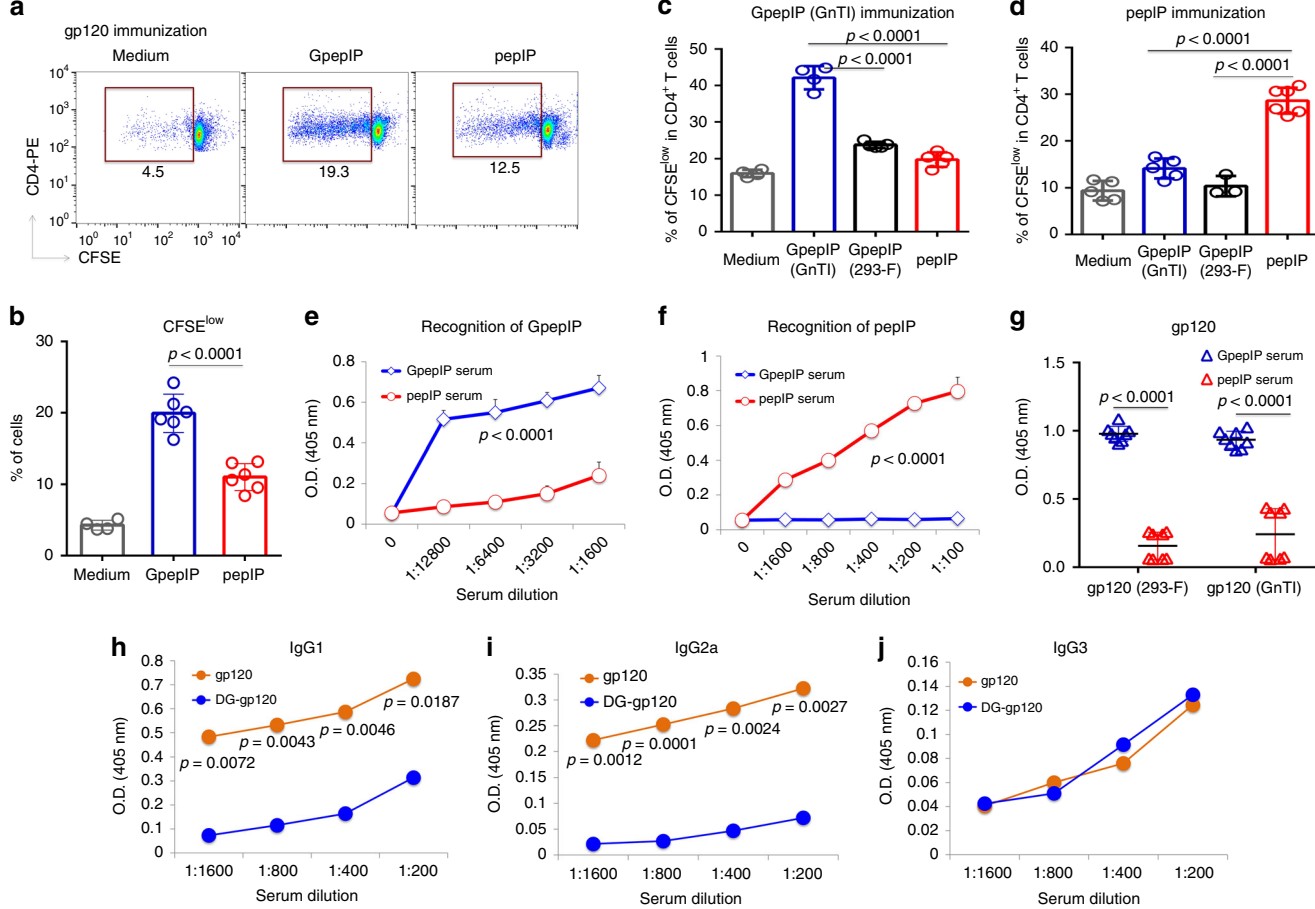

**Fig. 3 The glycopeptide epitope GpepIP elicits a glycan-dependent cellular and humoral immune response. a, b** CD4+ T cells obtained from mice immunized with gp120 were stimulated in vitro in the presence of mitomycin C-treated APCs pulsed with either GpepIP or pepIP, and T cell proliferation was examined by flow cytometry with use of CFSE fluorescence dilution through cell division. **c, d** CD4+ T cells obtained from mice immunized with GpepIP expressed in GnT1$^{-/-}$ cells **c** or with pepIP **d** were stimulated with GpepIP expressed in GnTI$^{-/-}$ or 293F cells or with pepIP in the presence of mitomycin C-treated APCs, and T cell proliferation was examined by CFSE flow cytometry. **e, f** Antisera from mice immunized with GpepIP or pepIP were titrated for IgG binding to immobilized GpepIP **e** or pepIP **f** by ELISA. **g** Antisera from GpepIP and pepIP immunization groups recognize gp120 expressed in 293-F and GnTI$^{-/-}$ cells differentially as measured by ELISA using a serum dilution 1:400. **h–j** Serum from mice immunized with GpepIP expressed in GnT1$^{-/-}$ cells were collected 7 days after booster immunization. As a control, serum from naïve mice was used as background. Titers of IgG1 **h**, IgG2a **i**, and IgG3 **j** for recognition of glycosylated gp120 or deglycosylated gp120 were measured by ELISA. Data was presented after subtracting background. Representative results are shown from one of three independent experiments performed. (mean ± s.d.). **b** medium $n = 4$; GpepIP and pepIP $n = 6$ independent experiments. **c** medium and GpepIP (GnTI) $n = 4$; GpepIP (293-F) and pepIP $n = 5$ independent experiments. $n = 4$ for **e** and **f**; $n = 8$ for **g**; $n = 2$ independent experiments for **h**, **i**, and **j**. P-values were determined using Student's two-sided t-tests. Source data are provided as a Source Data file.

and DG-gp120-specific IgG3 subclass appeared comparable (Fig. 3j). Consistent with results in Fig. 3e and f, GpepIP-immunized antisera showed markedly greater binding to GpepIP than pepIP in all three IgG subclasses (Supplementary Fig. 6c–e). Taken together, these results demonstrate that the GpepIP epitope induces glycopeptide-specific CD4+ T cell and antibody responses.

**GpepIP induces favored Th2 and Th17 differentiation.** To further characterize the CD4+ T cell responses induced by the non-conventional MHC ligand—GpepIP—and to obtain a gene expression profile of GpepIP-induced CD4+ T cells, we performed genome-wide transcriptomic analysis (RNA-seq) comparing GpepIP and pepIP-induced CD4+ T cell responses. Splenocytes from GnTI$^{-/-}$-expressed GpepIP and pepIP immunized mice were stimulated in vitro with GpepIP and pepIP, respectively, for 3 days. Using T cell activation marker CD69, we sorted out CD4+CD69+ T cell populations by flow cytometry whereas CD4+CD69$^-$ non-responding cells were used as control (Fig. 4a). High *cd4* gene expression was observed in all sorted groups and *cd69* and *Il2* were upregulated in GpepIP-stimulated and pepIP-stimulated groups compared to control (Supplementary Data 2 and 3). Hierarchical clustering of genes from each group revealed three distinct gene expression patterns with closer similarities between GpepIP and pepIP cells than with control (Fig. 4b). Comparing transcriptomes of GpepIP and control cells, we found that 3001 genes were differentially expressed (greater than twofold, $P < 0.05$) with 2460 genes upregulated, 541 genes downregulated and 12,711 genes unchanged (Supplementary Data 2 and Fig. 4c left). Gene ontology (GO) analysis showed a large fraction of differentially expressed genes (DEGs) in GpepIP group enriched in the biological processes associated with cellular process, metabolic process, response to stimulus, signaling, and immune system process (Supplementary Fig. 7a). Kyoto encyclopedia of genes and genomes (KEGG) pathways analysis of DEGs identified the enriched pathways were highly associated with immune functions, such as pathways in cancer, inflammatory bowel disease (IBD), hematopoietic cell lineage, cytokine–cytokine receptor interaction and leishmaniasis.

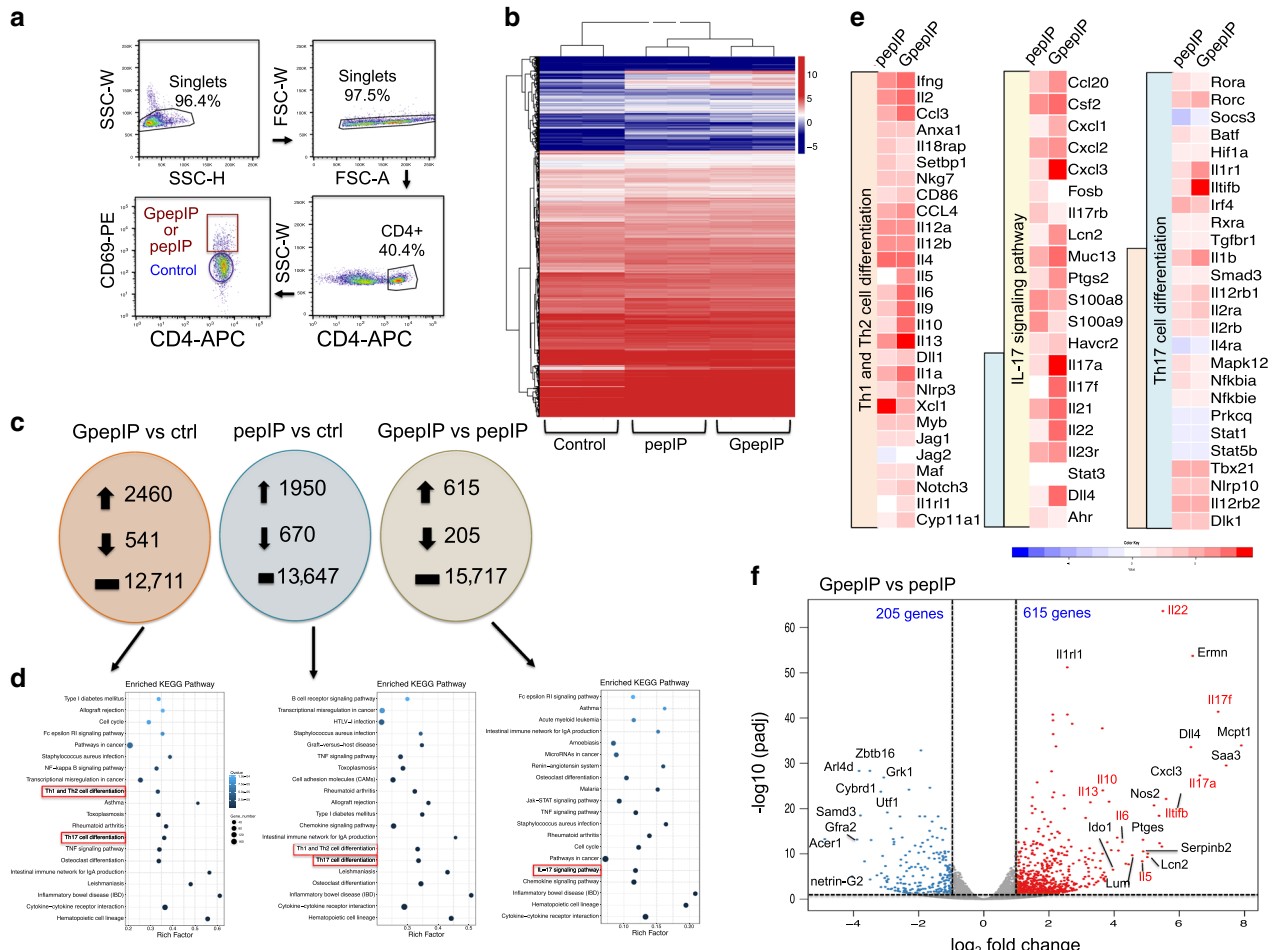

**Fig. 4 Transcriptomic analysis of GpepIP-stimulated and pepIP-stimulated CD4+ T cell populations. a** Gating strategy of sorting antigen-specific CD4+ T cell populations by flow cytometry is shown. Three weeks after the third immunization with GpepIP or pepIP, splenic and lymph node cells were isolated and stimulated in vitro with GpepIP or pepIP, respectively, for 3 days. CD4+CD69+ and CD4+CD69− T cell populations were then sorted by flow cytometry. **b** Dendrogram and hierarchical clustering heat map of genes from control, GpepIP, and pepIP populations. The blue and red bands indicate low and high gene expression quantity, respectively. The vertical distances between branches of the dendrogram represent the similarity of gene expression profiles between samples. Biological replicates showed the highest degree of correlation followed by GpepIP or pepIP stimulated populations. **c** The number of upregulated, downregulated, and unchanged genes in GpepIP and pepIP stimulated CD4+ T cells compared to control or to each other is shown. **d** KEGG pathway enrichment analysis of DEGs from each comparison. Point size indicates DEG number (the bigger dots refer to larger amount). Rich factor refers to the value of enrichment factor which is the quotient of the number of DEGs and total gene amount in that pathway. Pathways associated with Th cell differentiation were highlighted by red rectangles. **e** Heat map depicts the DEGs associated with Th1 and Th2 cell differentiation, IL-17-signaling pathway, and Th17 cell differentiation between GpepIP and pepIP specific CD4+ T cells with normalization to control. Heatmap colors represent the log2-fold change values relative to the control. **f** Volcano plot showing the gene signature of GpepIP compared to pepIP-specific CD4+ T cells. X-axis represents log2-transformed fold change. Y-axis represents −log10 transformed significance. Red points represent up-regulated DEGs. Blue points represent down-regulated DEGs. Gray points represent non-DEGs. Genes associated with Th cell differentiation were labeled and highlighted.

Particularly, GpepIP-specific CD4+ T cells were enriched for Th1, Th2, and Th17 cell differentiation compared to control (Fig. 4d left). Detailed examination of the top hits revealed a significant enrichment for genes associated with Th1 and Th2 signaling (*Ifng, Il2, Il2ra, Il1a, Il1r2, Il4, Il6, Il9, Il10, Il13, Mcpt1*), Th17 signaling (*Il17a, Il17f, Il22, Il23r, Iltifb, Dll4, Cxcl3*) and activated T cell co-stimulatory signaling (*cd44, cd63, cd83, Icos*) (Supplementary Fig. 7b).

Comparing transcriptomes of pepIP to control T cells, 2620 genes were differentially expressed with 1950 genes upregulated and 670 genes downregulated (Supplementary Data 3 and Fig. 4c middle). KEGG pathways analysis of DEGs of pepIP-specific CD4+ T cells also exhibited an enrichment of Th1, Th2, and Th17 cell differentiation (Fig. 4d middle). However, when comparing transcriptomes between GpepIP and pepIP specific CD4+ T cells, only 820 genes were differentially expressed (615

up-regulated and 205 down-regulated) (Supplementary Data 4 and Fig. 4c right). Notably, IL-17 signaling pathway was favorably enriched in GpepIP-induced CD4+ T cells (Fig. 4d right). A comprehensive list of all gene hits is included in Supplementary Data 4, and the transcription signatures associated with Th1 and Th2 cell differentiation, IL-17-signaling pathway and Th17 cell differentiation as noted in KEGG analysis were summarized as a heatmap normalizing to control (Fig. 4e). Prominent genes associated with Th1 differentiation appeared to have comparable expression levels in GpepIP and pepIP-specific CD4+ T cells, including *Ifng, Il2, Il18rap, Setbp1, Nkg7, cd86, Ccl4, Il12, Stat1*, and *Tbx21* (encoding T-bet) (Fig. 4e). Prominent genes associated with Th2 differentiation, however, were highly upregulated in GpepIP compared to pepIP induced CD4+ T cells, such as *Il5, Il6, Il9, Il10, Il13, Nlrp3, Il1rl1, Cyp11a1, Mcpt1*, and *Serpinb2* (Fig. 4e, f). Of note, produced by both Th2 and follicular helper T (Tfh)

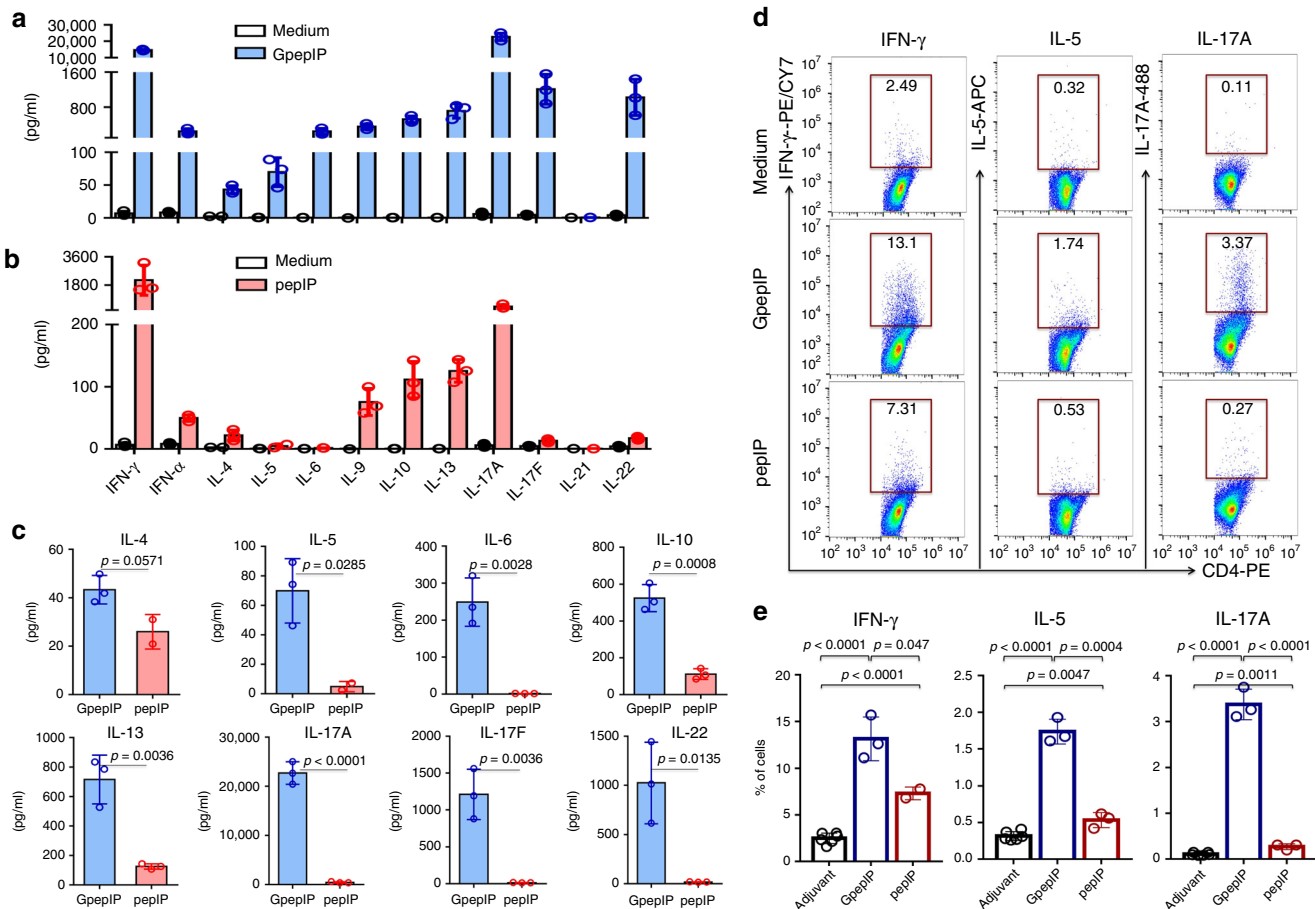

**Fig. 5 Cytokine profile of GpepIP and pepIP stimulation.** Splenic and lymph node cells isolated from GpepIP or pepIP immunized mice were stimulated with GpepIP or pepIP, respectively, for 5 days. Th-cell-related cytokines in the supernatants from GpepIP **a** or pepIP **b** stimulation compared to no stimulation (medium) were analyzed by a multiplex-based assay. **c** Production of cytokines associated with Th2 (IL4, IL-5, IL-6, IL-10, and IL-13) and Th17 (IL-17A, IL-17F, and IL-22) was examined in GpepIP-stimulated and pepIP-stimulated groups. **d, e** Cells in **a** and **b** were stimulated with GpepIP or pepIP or in medium for 3 days. Cytokines IFN-γ, IL-5, and IL-17A on CD4+ T cells were assessed by intracellular cytokine staining and flow cytometry. Representative results are shown from one of two independent experiments performed. (mean ± s.d.). **a–c** $n = 3$ independent experiments. **e** $n = 6$ for medium; $n = 3$ independent experiments for GpepIP and pepIP. $P$-values were determined using Student's two-sided $t$-tests. Source data are provided as a Source Data file.

cells[26], the expression of *Il4* showed no difference between GpepIP and pepIP (Fig. 4e). Strikingly, the expression of genes associated with Th17 signature was remarkably elevated in GpepIP-specific CD4+ T cells, including *Il17a, Il17f, Il22, Iltifb, Il23r, Rorc* (encoding RORγt), *Cxcl1, Cxcl3, Lcn2, Dll4, Il1r1, Saa3, Nos2, Ptges, Lum,* and *Ido1* (Fig. 4e, f), indicating a robust Th17 differentiation elicited by GpepIP.

The Th cell differentiation status of GpepIP and pepIP specific CD4+ T cells was further validated at the protein level by assessing Th1, Th2, and Th17 signature cytokines in T cell cultured supernatant. After a 5-day GpepIP or pepIP antigen stimulation of T cells from GpepIP or pepIP immunized mice, respectively, supernatants were harvested for a multiplex-based cytokine measurement. Consistent with RNA-seq data, both GpepIP and pepIP stimulated supernatants displayed significantly increased Th1 and Th2 cytokines production compared to medium group (Fig. 5a, b). Despite the Th2 enrichment in both GpepIP and pepIP groups, signature cytokines after GpepIP stimulation showed markedly augmented expression, such as IL-5, IL-6, IL-10, and IL-13 (Fig. 5c). Yet, similar IL-4 expression was observed in both groups (Fig. 5c). Although pepIP stimulation induced increased IL-17A production over medium alone, the

extent of its expression was strikingly lower than GpepIP groups (Fig. 5c). Additionally, the expression levels of two other Th17-related cytokines IL-17F and IL-22 were substantially lower in pepIP than GpepIP group (Fig. 5c).

In addition, representative Th1, Th2, and Th17 cytokines IFN-γ, IL-5, and IL-17A were evaluated by intracellular cytokine staining combined with flow cytometry. The results showed that after 3 days of antigen stimulation, GpepIP-specific CD4+ T cells displayed an overall higher expression of all three cytokines with greater enhancement in Th2 and Th17 cytokines (Fig. 5d, e). The higher IFN-γ production in GpepIP than pepIP specific CD4+ T cells detected at the protein level (Fig. 5) are not in accord with mRNA results (Fig. 4); this is likely at least partially due to different time points of sample collection. When cytokines were detected on day 5 after antigen stimulation, IFN-γ production was comparable between GpepIP and pepIP groups, albeit IL-5 and IL-17A were still higher in GpepIP-specific CD4+ T cells (Supplementary Fig. 8a, b). Taken together, transcriptomic analysis and cytokine assessment results demonstrate that the pepIP epitope activates mainly Th1 polarized CD4+ T cells whereas glycosylation on this epitope (GpepIP) induces a distinct and strong Th2 and Th17 enrichment.

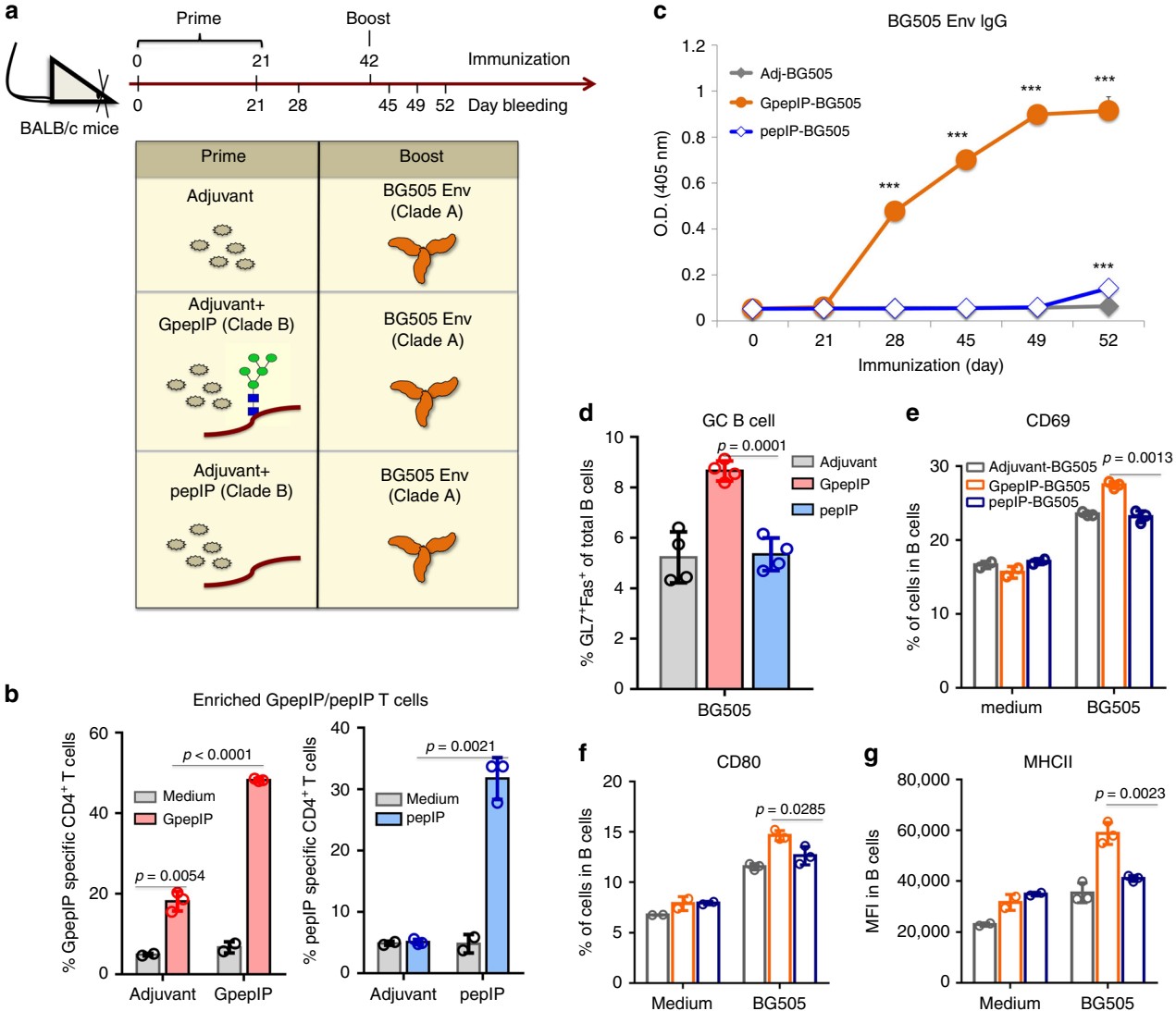

**Fig. 6 GpepIP-specific CD4+ T cells exhibit high potency on helping humoral immune responses to HIV-1 trimer. a** Immunization scheme. BALB/c mice were primed twice by subcutaneous injection of GpepIP or pepIP emulsified in Freund's adjuvant or of adjuvant alone. Three weeks later, all groups were immunized with the clade A BG505 gp140 NFL trimer emulsified in incomplete Freund's adjuvant. Sera were collected at the indicated time points. Mice were euthanized 10 days after trimer immunization. **b** Splenic and lymph node cells were isolated and stimulated with GpepIP or pepIP in vitro for 5 days. T cell proliferation by CFSE dilution was measured by flow cytometry. **c** BG505-specific IgG production was examined in all three groups across the indicated time points by ELISA using a serum dilution 1:100. **d** GC response, defined by the percentage of GL7+Fas+ B cells among B220+ B cells, was evaluated in all three groups on day 10 after trimer immunization by flow cytometry. **e–g** Expression levels of activation marker CD69 **e**, CD80 **f**, and MHCII **g** were detected on splenic B cells after in vitro stimulation with the BG505 trimer for 3 days. MFI mean fluorescence intensity. Representative results are shown from one of three independent experiments performed. (mean ± s.d.). **b** medium $n = 2$; GpepIP or pepIP $n = 3$ independent experiments. **d** $n = 4$ independent experiments. **e–g** medium $n = 2$; BG505 $n = 3$ independent experiments. *P*-values were determined using Student's two-sided *t*-tests. Source data are provided as a Source Data file.

**GpepIP CD4+ T cells help HIV-1 Env humoral immune responses.** CD4+ T cells are essential mediators in helping B cells and antibody responses[26]. We next sought to assess whether GpepIP-induced CD4+ T cell facilitates antibody responses to Env trimer—a more immunologically relevant challenge. The B cell-help by pepIP-specific CD4+ T cells was also investigated to determine the specificity and potency of glycopeptide T cell epitopes over peptide epitopes. BALB/c mice were primed with GnTI−/−-expressed GpepIP or pepIP emulsified in Freund's adjuvant or with adjuvant alone; all groups were boosted with a clade A BG505 gp140 native flexibly linked (NFL) trimer[16]. Antisera were collected at specific time points (Fig. 6a). GpepIP-specific and pepIP-specific CD4+ T cells were induced in

GpepIP-primed and pepIP-primed mice, respectively, compared to adjuvant group, as evidenced by significantly greater T cell proliferation in the GpepIP-primed and pepIP-primed groups after in vitro stimulation with GpepIP and pepIP (Fig. 6b). The stimulation of T cells by GpepIP in the adjuvant-primed group is likely due to the response to the shared glycopeptide/glycan epitopes from Env trimer after BG505 booster immunization (Fig. 6b). With distinct antigen-specific CD4+ T cell help, Env trimer-specific IgG production was compared in these mice across the indicated time points (Fig. 6c). Substantially higher BG505-specific IgG titers were observed in mice primed with GpepIP than with adjuvant or pepIP ($P < 0.0001$) (Fig. 6c). The help from pepIP priming was different from adjuvant priming

only at the experimental endpoint (Fig. 6c). Of note, GpepIP priming yielded antibody responses against the BG505 Env trimer even before the trimer was given (day 28); this is most likely due to shared glycopeptide/glycan antibody epitopes between GpepIP and Env trimer (Fig. 6c).

To directly assess the role of GpepIP-specific CD4+ T cells in helping HIV Env trimer-specific humoral immune responses, we performed CD4+ T cell adoptive transfer and CD4+ T cell depletion experiments. CD4+ T cells were purified from donor mice immunized with GpepIP, pepIP, or adjuvant and adoptively transferred into naïve recipient mice followed by BG505 trimer immunization. Sera were collected 7 days after trimer boost and tested for BG505-specific IgG and IgM by ELISA. While Env trimer-specific IgM production was comparable in all three groups (Supplementary Fig. 9b), Env trimer-specific IgG titer was significantly higher in mice that received CD4+ T cells from GpepIP-immunized donor mice than that received CD4+ T cells from pepIP-immunized and adjuvant-immunized mice (Supplementary Fig. 9a). Furthermore, mice were immunized as in scheme Fig. 6a. Their CD4+ T cells were depleted during BG505 trimer immunization by injecting anti-CD4 mAb to diminish T cell help. In agreement with previous results (Fig. 6c), GpepIP-primed mice injected with isotype mAb showed significantly elevated BG505 IgG production than adjuvant and pepIP-primed mice (Supplementary Fig. 9c). However, the helping effect from GpepIP priming was abolished after CD4+ T cell depletion (Supplementary Fig. 9c). The BG505 IgM levels were not altered among groups (Supplementary Fig. 9d). These data demonstrate that HIV Env trimer-specific booster humoral immune response is mediated by GpepIP-specific CD4+ T cell priming.

In agreement with serum IgG titers, Env trimer-specific B-cell responses were enhanced in GpepIP-primed mice as well. Compared to adjuvant and pepIP, GpepIP primary immunization induced superior germinal center (GC) response, defined by a significantly increased percentage of GL7+Fas+ B cells (Fig. 6d). Moreover, the expression of B cell activation markers CD69, CD80, and MHCII was higher among splenic B cells isolated from GpepIP primed mice after in vitro stimulation with BG505 trimer (Fig. 6e–g). These results indicate that GpepIP-induced CD4+ T cells provide effective and potent help for Env trimer antibody responses compared to pepIP-specific CD4+ T cells.

**GpepIP CD4+ T cells enhance functional antibody responses.** With the observation of GpepIP eliciting HIV-1 envelope-specific humoral immune responses, we next sought to determine the contribution of GpepIP-induced CD4+ T cells to the functional antibody development. We applied an immunization regimen with three prime immunizations of adjuvant, GpepIP, or pepIP, followed by three boost immunization of BG505 to improve GpepIP-induced CD4+ T cell help and Env trimer-specific IgG responses (Fig. 7a). Env trimer-specific booster antibody responses were assessed by ELISA (Fig. 7b). In agreement with previous results (Fig. 6c), GpepIP priming significantly increased BG505 IgG production than adjuvant or pepIP-priming groups after first trimer boost (Post 1). The IgG titers of GpepIP-priming and pepIP-priming groups were similar after second trimer boost (Post 2), while both were significantly higher than adjuvant group. After third BG505 immunization, all three groups had comparable BG505 IgG titers (Post 3). We then evaluated the quality and function of antisera after three trimer immunizations of all tested groups. Different effector Th subsets have been identified to promote antibodies class switching[27,28]. Antisera from all three groups after BG505 boost for three times—with equivalent BG505 IgG titers—were tested for their IgG subclasses. Among all, IgG1 subclass represented the predominant antibody subclass in all immunization groups without

significant difference between groups. While pepIP priming induced higher BG505 IgG2a subclass, GpepIP-specific CD4+ T cells preferentially helped BG505 IgG class switching towards IgG2b and IgG3 (Fig. 7c).

The neutralizing activity of these sera was then assessed via a TZM-bl cell-based neutralization assays[29,30]. Antisera from all tested groups after three BG505 booster immunizations were assayed against tier 1 and tier 2 HIV-1 viruses. Compared to trimer immunization only (adjuvant), we detected clear neutralization of the tier 1A virus MN.3 in all sera samples from GpepIP pre-immunization group (Fig. 7d). Notably, sera from pepIP pre-immunization did not also show neutralizing activities. Antisera from this pilot assay did not show broad neutralizing breadth. On the other hand, there is emerging evidence indicating that functional non-neutralizing antibodies provide effective protection against HIV-1 infection[14]. An antibody-mediated immunogen-uptake assay was established based on our previous method[21] to evaluate antibody function. With equivalent Env trimer-specific IgG titers (Supplementary Fig. 9e), pre-incubation of fluorophore-labeled BG505 with antisera from GpepIP-primed and BG505-boosted mice significantly promoted the BG505 uptake by BMDCs compared to incubation with no serum, and more importantly, compared to sera from adjuvant-primed or pepIP-primed mice (Fig. 7e). Taken together, these data indicate that GpepIP epitope, but not the naked peptide epitope, elicits a repertoire of CD4+ T cells that help HIV-1 envelope-specific-B cells to produce enhanced, protective, and functional antibodies.

## Discussion

Identification and characterization of bNAbs (several of which recognize the glycan shield of gp120[2,3,6–8]) and of non-neutralizing polyfunctional antibodies[14,31] from HIV-1-infected individuals have buoyed hopes that inducing the production of protective antibodies through immunization can halt the spread of infection. Efforts toward intelligent implementation of this strategy have benefited from a deep understanding of B-cell receptor/envelope recognition[14,32–35].

In this study, we addressed the other critical direction of HIV-1 research: elucidation of gp120-induced T cell activation mechanisms underlying elicitation of Env trimer-specific humoral immune responses. Recent studies have demonstrated critical roles for CD4+ helper T cells in driving antibody subclass switching, affinity maturation, and effector function of antibodies to HIV-1[26,36]. Particularly, generation of bNAbs necessitate affinity maturation and somatic mutations, involving CD4+ T cell help[36]. The RV144 vaccine trial showed a correlation of CD4+ T cell response with protection, indicating the contribution of helper T cells in protective antibody production[37].

Past efforts to identify gp120 T cell epitopes focused solely on naked peptide epitopes[38]. By using overlapping peptides spanning the entire gp120, it was suggested that the promiscuously immunodominant gp120 T cell epitopes were clustered forming CD4+ T cell epitope "hot spots"[39,40]. However, gp120 proteins are heavily glycosylated with various numbers (from 23 to 26) of N-linked glycosylation sites across the protein region[41,42]. It has been shown that N-linked glycans on gp120 influence CD4+ T cell responses by modulating antigen processing[43] or epitope generation[44]. More importantly, within APCs, gp120 glycan moieties survive the antigen processing, yielding glycopeptide epitopes presented by MHCII to CD4+ T cells. Therefore, gp120 glycan shield will significantly skew CD4+ T cell epitope "hot spots". Thus, it is reasonable to evaluate the importance of gp120 glycopeptide-specific CD4+ T cells. Additionally, the view that carbohydrates serve directly as non-conventional epitopes to induce CD4+ T cell responses have received increasing appreciation[19,20,23,45]. Here, we isolated and

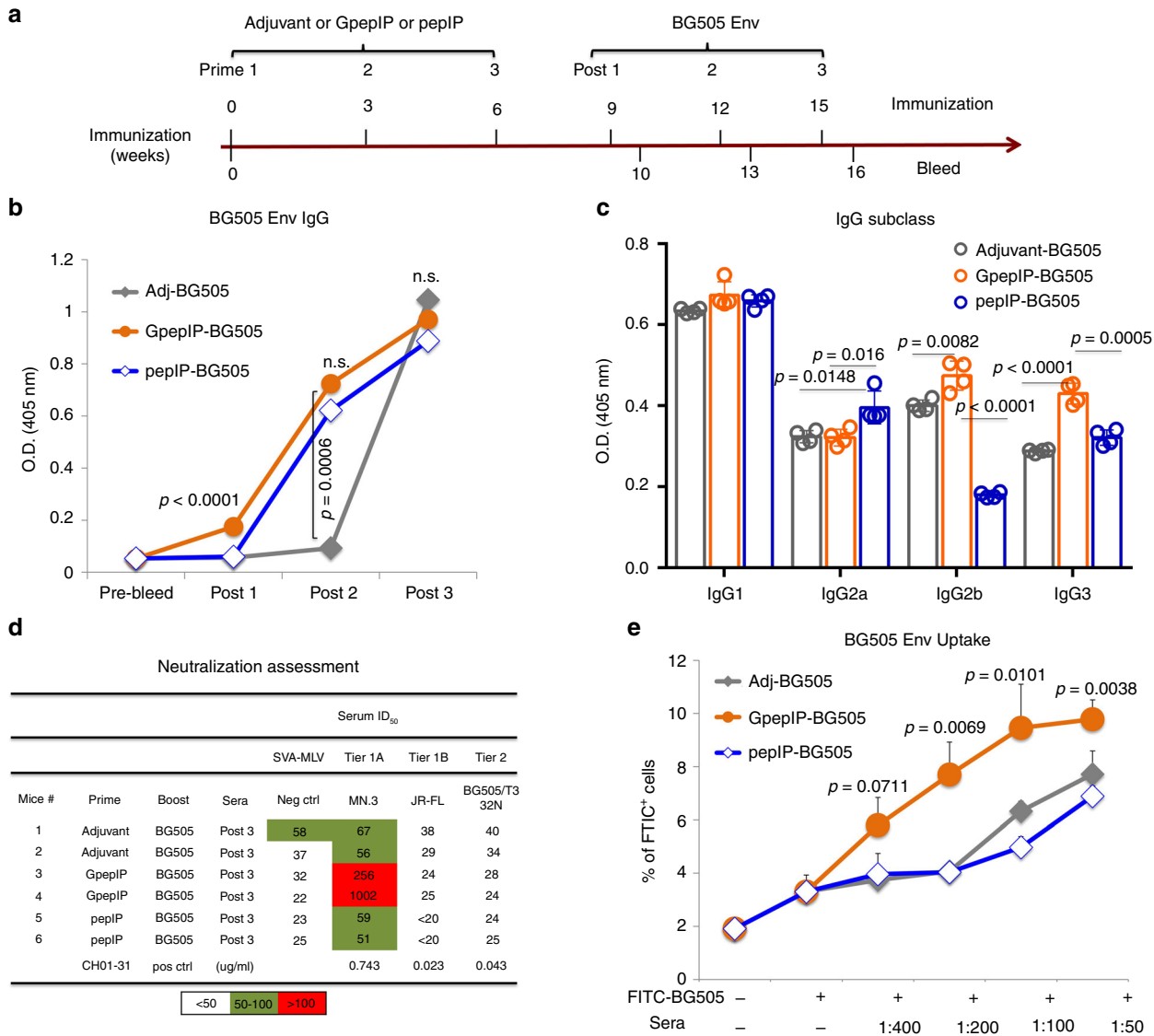

**Fig. 7 GpepIP primary immunization prior to trimer immunization elicits functional antibody production. a** Immunization scheme. BALB/c mice were primed three times (with a 3-week interval) by subcutaneous injection of GpepIP or pepIP emulsified in Freund's adjuvant or of adjuvant alone. Subsequently, all groups were immunized with the clade A BG505 gp140 NFL trimer adjuvanted with Alum for three times (with a 3-week interval). Sera were collected 7 days after each trimer immunization (post 1–3). **b** BG505-specific IgG production was examined in all three groups post each trimer immunization by ELISA using a serum dilution 1:400. **c** Antisera from all three groups after the third trimer immunizations were analyzed for IgG subclass switching by ELISA using a serum dilution 1:1600. **d** The neutralizing activity (neutralization 50% inhibitory dilution (ID$_{50}$)) of antisera from adjuvant, GpepIP, and pepIP primary followed by three BG505 booster immunizations were tested against tier 1 and tier 2 HIV-1 viruses via a TZM-bl cell-based neutralization assays. MLV-pseudotyped virus was used as negative control for non-HIV-1-specific inhibitory activity in the assay. Antibody CH01-31 was used as positive control (shown as antibody concentration). **e** BG505 was chemically labeled with fluorescein isothiocyanate (FITC) and incubated with BMDCs at 37 °C for 2 h. Cells were then collected and antigen uptake was measured by flow cytometry. To evaluate antisera for their function, fluorophore-labeled BG505 was pre-incubated with antisera used in **e** at different dilutions before adding into BMDCs. The uptake rate is defined as FITC-positive cells compared to no BG505 (medium) group. Representative results are shown from one of three independent experiments performed. (mean ± s.d.). **b** n = 6 for pre-bleed; n = 2 independent experiments for post 1–3. n = 4 and 3 independent experiments for **c** and **e**, respectively. P-values were determined using Student's two-sided t-tests. Source data are provided as a Source Data file.

characterized an MHCII-presented glycopeptide epitope (GpepIP) eliciting T cell-mediated humoral immune responses to the HIV-1 envelope glycoprotein. Our data demonstrate that GpepIP glycans directly participate in CD4$^+$ T cell recognition, evidenced by: (1) loss of glycans impairs the T cell response to a glycopeptide epitope; (2) alteration of glycan structure dampens the T cell response; and (3) addition of glycans blocks the T cell response to a peptide epitope. Future studies are required to reveal the recognition specificities of GpepIP-specific CD4$^+$ T cells, such as evaluating alternative

peptides containing the same glycan structure. While the glycan portion of GpepIP is a product of host glycosylation machinery, it is still antigenic. Self-recognizing T cells are not expected to be a concern due to the fact that the dominant oligo/high-mannose structures on HIV-1 Env or GpepIP are markedly distinct from predominant host (mammalian) N-glycans[42,46] and very few glycan-targeting bNAbs are autoreactive but still protective[47].

Previous studies have reported that carbohydrate-specific signaling shapes Th cell differentiation through modulating the

crosstalk between APCs and CD4[+] T cells[48–50]. In this study, we identified Th cell differentiation programming upon glycopeptide epitope recognition. The comprehensive RNA-seq analysis and Th-associated cytokine profiles demonstrate that a peptide epitope (pepIP) induces a Th1 dominant immune response, whereas recognition of a glycopeptide epitope (GpepIP) by CD4[+] T cells drives the induction of Th differentiation towards Th2 and Th17 features. Further investigation is needed to elucidate the mechanisms underlying regulation of CD4[+] T cell differentiation and function by their glycan recognition.

Functionally, GpepIP epitope shows greater effectiveness and potency in eliciting T cell help for HIV-1 envelope humoral immune response than pepIP epitope. Further, GpepIP (derived from clade B) induced CD4[+] T cells display broader help breadth for a heterologous Env trimer BG505 (clade A). The GpepIP region is highly variable across gp120s from different clades[51]. The equivalent peptide sequence and glycans from BG505 are significantly different from GpepIP[42]. We have shown that GpepIP-specific CD4[+] T cells cross-react with shared glycopeptide and/or glycan epitopes generated after BG505 processing in APCs. However, it is still important to investigate whether the equivalent glycopeptide from BG505 also binds to MHCII and stimulates the GpepIP-induced CD4[+] T cells. Tfh cells are CD4[+] T cells specialized in helping B cell-mediated immunity and antibody responses[52]. However, our data indicates that neither GpepIP nor pepIP-specific CD4[+] T cells are Tfh cells as the expression of genes associated with Tfh cell programing, such as *Bcl6, Pdcd1* (encoding PD-1), *Sh2d1a* (encoding SLAM-associated protein (SAP)), and *Btla* showed no difference from control group; and minimal IL-21 production was detected. The superior antibody responses by GpepIP over pepIP is most likely due to GpepIP stimulating more effective Th2 and Th17 responses than the pepIP[27,53,54]. GpepIP elicits substantial antibody response targeting gp120 glycan-epitopes shared by immunogens across clades, further contributing to GpepIP-specific CD4[+] T cells' potency. Analyses of RV144 vaccine trial identified a unique immune response profile, marked by V2-specific IgG3 antibodies and IL-13 signature from envelope-stimulated PBMC supernatant[12,55], suggesting the functional potential of GpepIP elicited Th2 and IgG3 responses.

Importantly, as a proof-of-principle for driving functional antibody responses through eliciting glycopeptide-specific helper T cell activation, we demonstrated that GpepIP primary immunization followed by BG505 booster immunization resulted in tier 1 neutralizing antibody development, while BG505 booster immunization alone (adjuvant pre-immunization) and pepIP pre-immunization did not. With equivalent IgG titers, BG505 antibodies from GpepIP-primed mice have a greater functional capability to mediate antigen uptake by APCs than adjuvant-primed or pepIP-primed mice. It is important to compare GpepIP epitope with a naturally processed non-glycosylated peptide in helping Env antibody response as part of future studies. Adaptive immune mechanisms induced by glycopeptides of the HIV-1 envelope glycoprotein described here offer an innovative approach for future clinical development of knowledge-based, functional HIV-1 vaccines.

## Methods
**Mice**. Eight-week-old female BALB/c mice were obtained from The Jackson Laboratory (Bar Harbor, ME) and housed in the Center for Molecular Medicine Rodent Vivarium at the University of Georgia with housing condition of 20 °C temperature and 60% humidity. Mice were kept in microisolator cages and handled under a laminar flow hood. All mouse experiments were conducted in compliance with the University of Georgia Institutional Animal Care and Use Committee under an approved animal use protocol.

**Antigen production**. The codon-optimized pSyn gp120 plasmid encoding JR-FL (clade B) gp120 was obtained from Dr. Eun-Chung Park and Dr. Brian Seed

through the NIH AIDS Reagent Program, Division of AIDS, NIAID, NIH[56,57]. The HIV-1 BG505 gp140 NFL expression vector was obtained from Dr. Richard T. Wyatt through the NIH AIDS Reagent Program, Division of AIDS, NIAID, NIH[58]. Both of them were expressed in a serum-free medium by transient transfection of wild-type FreeStyle™ 293-F suspension cells (Life Technologies); gp120 and BG505 trimer were purified from supernatant by affinity chromatography using *Galanthus nivalis* lectin-agarose (Vector Labs, Burlingame, CA) and then further purified on a Superdex S200 size exclusion column (Bio-Rad) to isolate gp120 and trimer fraction as described previously[21,59]. gp120 was also expressed and purified from HEK293S GnTI[−/−] cells (ATCC).

To generate a glycopeptide pool, gp120 was digested by sequencing-grade Endoproteinase Glu-C (Promega) in ammonium bicarbonate buffer. Digested products were separated on a Superdex Peptide 10/300 GL column (GE Healthcare) in order to selectively isolate glycopeptides from non-glycosylated peptides. Glycopeptide-containing fractions were identified and pooled together on the basis of biotinylated ConA (Vector Labs) reactivity in a lectin dot blot. Pooled glycopeptides were desalted on a HiPrep 26/10 Desalting column (GE Healthcare) and lyophilized overnight.

The gp120 GpepIP expression vector was generated by cloning of the synthetic nucleotide sequence (GENEWIZ) that codes for the signal sequence-6xHistidine-GpepIP peptide into the pGEn2 restriction-site mammalian expression vector (provided by Dr. Kelley Moremen, University of Georgia) via NotI and BamHI restriction sites. GpepIP expression vector containing a His tag was generated. Recombinant GpepIP was expressed in transiently transfected 293-F cells or HEK293S GnTI[−/−] cells (ATCC) as described previously[21]. GpepIP was purified by passage of supernatant through Ni[2+]-NTA resin (G-bioscience) at 4 °C, washing with buffer containing 10 mM imidazole (G-bioscience) and 0.2 M NaCl, and elution with 0.3 M imidazole. The elution was collected, desalted, and lyophilized for long-term storage. The purity of glycopeptides was evaluated by SDS–PAGE and silver staining (Pierce). Concentrations of GpepIP and pepIP were determined by the quantitative fluorometric peptide assay (Pierce) according to the manufacturer's protocol. All peptides, including pepIP, OVA peptide 323–339, and scrambled OVA peptide, were purchased from GenScript (Piscataway, NJ).

For removal of N-linked glycans, PNGase F (a gift from Dr. Kelley Moremen, University of Georgia)[60] was applied under denaturing conditions as previously described[21]. In brief, glycoproteins or glycopeptides were boiled in 20 mM sodium phosphate buffer (pH 7.5) containing 0.05% SDS and 50 mM β-mercaptoethanol. PNGase F was added to the protein at a ratio of 1:10; overnight incubation at 37 °C followed.

**Immunization**. Groups of BALB/c mice were immunized on days 0 and 21 with 10 μg of gp120 emulsified in Freund's adjuvant (Thermo Scientific) by subcutaneous injection dorsal to the base of the tail. For glycopeptide or peptide immunization, 30 μg of antigen was used. During prime-boost immunization experiments, mice were bled from the tail vein 7 days after boosting.

For evaluation of the role of GpepIP-specific and pepIP-specific CD4[+] T cells in selectively eliciting trimer antibody production, mice were primed twice (with a 3-week interval) with either GpepIP emulsified in Freund's adjuvant or pepIP or adjuvant alone. 21 days after the second priming, both groups of mice were boosted with 20 μg of the BG505 trimer emulsified in incomplete Freund's adjuvant. Sera were collected at the indicated time points.

For evaluation of functional antibody responses, BALB/c mice were primed three times (with a 3-week interval) by subcutaneous injection of GpepIP or pepIP emulsified in Freund's adjuvant or of adjuvant alone. Three weeks later, all groups were boosted by intraperitoneal (i.p.) injection of 20 μg of the clade A BG505 gp140 NFL trimer adjuvanted with 2% alhydrogel (Invivogen) for three times (with a 3-week interval). Sera were collected 7 days after each trimer boost.

**T cell proliferation, B cell activation, and flow cytometry**. Single cell suspensions from lymph nodes and spleens were generated by gently grinding the tissue between sterile glass slides. CD4[+] T cells were isolated from lymph nodes of mice 3 weeks after booster immunization with a mouse CD4 T lymphocyte enrichment set (BD Biosciences) according to the manufacturer's protocol. CD4[+] T cells were stimulated in vitro in the presence of mitomycin C-treated (25 μg/ml, Sigma-Aldrich) splenic mononuclear cells pulsed with 10 μg/ml of indicated antigens (20 μg/ml for glycopeptide and peptide) for 5 days. In some experiments, total splenic and lymph node cells were used. For CFSE labeling, CD4[+] T cells were incubated with 2 μM CFSE solution (Sigma-Aldrich) at 37 °C for 8 min before stimulation. CFSE dilution was measured by flow cytometry as an indication of the T cell proliferation rate. To evaluate the CD4[+] T cells response to GpepIP and pepIP is MHCII-restricted, CD4[+] T cells from GpepIP or pepIP immunized mice were stimulated with GpepIP or pepIP, respectively, in the presence of anti-mouse I-A/I-E antibody (Biolegend, clone M5/114.15.2) or Rat IgG2b isotype control (Biolegend, clone RTK4530) at the indicated concentration.

To evaluate CD4[+] T cell-mediated B cell responses, mice were sacrificed 10 days after trimer booster imunization. GC response was assessed by the expression of GL7 and Fas among splenic B cells by flow cytometry. Splenocytes were stimulated in vitro with 10 μg/ml of BG505 trimer for 3 days to examine the expression of activation markers CD69, CD80, and MHCII by flow cytometry.

Surface staining of cell suspensions was performed by incubation of indicated mAbs for 30 min in PBS containing 0.1% BSA and 0.02% NaN₃ at 4 °C in the dark[61]. The following fluorophore-conjugated antibodies used in flow cytometry detection were purchased from BioLegend: anti-mouse CD4 (dilution 1:3000, clone GK1.5), anti-mouse CD4 (dilution 1:3000, clone RM4-4), anti-mouse/human CD45R/B220 (dilution 1:2000, clone RA3-6B2), anti-mouse/human GL7 (dilution 1:400, clone GL7), anti-mouse CD95 (Fas) (dilution 1:400, clone SA367H8), anti-mouse CD69 (dilution 1:1600, clone H1.2F3), anti-mouse CD80 (dilution 1:1000, clone 16-10A1), and anti-mouse I-A/I-E (dilution 1:2000, clone M5/114.15.2). Fluorescent signal was analyzed on CytoFLEX (Beckman Coulter, Hialeah, FL). Data were analyzed with FlowJo V10.1 software (Tree Star, Inc., Ashland, OR).

**CD4⁺ T cells adoptive transfer and CD4⁺ T cells depletion**. BALB/c mice were immunized twice (with a 3-week interval) by subcutaneous injection of GpepIP or pepIP emulsified in Freund's adjuvant or of adjuvant alone. 5 days after boost immunization, CD4⁺ T cells were isolated from spleen and lymph nodes from each group. $1 \times 10^7$ CD4⁺ T cells were adoptively transferred to recipient mice through tail-vein injections in PBS. The recipient mice were then immunized with 20 μg of the clade A BG505 gp140 NFL trimer emulsified in Freund's adjuvant 1 day after adoptive transfer. In addition, BALB/c mice were primed twice with GpepIP or pepIP emulsified in Freund's adjuvant or of adjuvant alone. Three weeks later, all groups were immunized with the BG505 trimer emulsified in incomplete Freund's adjuvant. To deplete CD4⁺ T cells, mice were i.p. injected with 250 μg either anti-mouse CD4 (Biolegend, clone GK1.5) or Rat IgG2b isotype control (Biolegend, clone RTK4530) mAbs in 500 μl PBS 3 days and 1 day before, and 2 days after trimer immunization. Depletion was verified by flow cytometry 3 days after trimer immunization by staining peripheral blood lymphocytes with anti-mouse CD4 antibody (Biolegend, dilution 1:3000, clone RM4-4). The level of depletion had >98% efficiency. Sera were collected 7 days after trimer boost immunization. BG505-specific IgG and IgM production were examined by ELISA.

**Cytokine analysis**. After a 5-day GpepIP or pepIP antigen stimulation of T cells from GpepIP or pepIP immunized mice, supernatants were harvested, and the concentrations of Th cell-associated cytokines were assessed using LEGENDplex mouse Th cytokine immunoassay kit (BioLegend) according to the manufacturer's instructions. Data were analyzed with LEGENDplex Software V7 software (Biolegend).

For intracellular cytokine staining, splenic and lymph node cells were isolated from GpepIP and pepIP immunized mice, and stimulated in vitro with GpepIP and pepIP, respectively, for 3 or 5 days. GolgiPlug (BD Biosciences) was added at the recommended concentration in the last 6 h of stimulation. Cells were collected and stained for surface marker CD4 (Biolegend). Intracellular staining of IFN-γ, IL-5, and IL-17A was performed following fixation and permeabilization of stimulated cells with Cytofix/Cytoperm solution (BD Pharmingen) using the following antibodies: anti-mouse IFN-γ (dilution 1:200, clone XMG1.2), anti-mouse/human IL-5 (dilution 1:200, clone TRFK5), and anti-mouse IL-17A (dilution 1:400, clone TC11-18H10.1). All antibodies and isotype fluorescence-conjugated antibodies were purchased from Biolegend. Samples were analyzed on CytoFLEX (Beckman Coulter, Hialeah, FL). Data were analyzed with FlowJo V10.1 software (Tree Star, Inc., Ashland, OR).

**Enzyme-linked immunosorbent assay (ELISA)**. Cytokine production resulting from T cell stimulation was measured by ELISA. In brief, 96-well plates (Costar) were coated overnight with antibody to IFN-γ or IL-4 (1:1000 dilution; Biolegend) and blocked with 1% BSA/PBS. Plates were washed with 0.05% PBS-Tween and incubated with cell supernatants for 2 h at room temperature. After washing, biotinylated detection antibodies to IFN-γ or IL-4 (1:1000 dilution; Biolegend) were added for 2 h at room temperature, after which HRP-conjugated Avidin (1:1000 dilution; Biolegend) was added for 1 h at room temperature. Plates were developed with a 3,3′,5,5′-tetramethylbenzidine substrate (TMB; Biolegend), and development was stopped with 2 N $H_2SO_4$. The optical densities were determined at 450 nm with a microplate reader (Synergy H1, Bio-Tek).

Antigen-specific antibodies in sera were measured by ELISA. Briefly, 96-well plate was coated with anti-gp120 mAb D7324 (dilution 1:200, Aalto Bioreagents, Dublin, Ireland) to capture gp120 and BG505 trimer. GpepIP and pepIP (5 μg/ml) were directly coated onto 96-well plates. After washing and blocking steps, serial dilutions of serum from immunized mice were added to the plate for 2 h at room temperature. Total IgG was detected by alkaline phosphatase (AP)-conjugated anti-mouse IgG (dilution 1:2000, Southern Biotech) coupled with AP substrate (Sigma-Aldrich). The optical densities were determined at 405 nm. IgG subclasses were detected by HRP-conjugated anti-mouse IgG1, IgG2a, IgG2b, and IgG3 (dilution 1:10,000, Abcam) and TMB substrate as described above.

To clarify epitope recognition of antisera from immunized mice, inhibition ELISA was performed. Sera diluted 1:1600 were pre-incubated with different concentrations of inhibitors before being added to a plate pre-coated with gp120 protein (5 μg/ml). IgG production was measured as described above.

**Purification of MHCII-bound glycopeptides**. BMDCs were generated after GM-CSF (PeproTech) induction for 8 days as described previously[21] and were incubated with gp120 protein (250 μg/ml) at 37 °C for 18 h. Cell lysate was prepared with the following lysis buffer (pH 8.0): 20 mM Tris–HCl, 137 mM NaCl, 1% Nonidet P-40 (NP-40), and 2 mM EDTA with a protease inhibitor cocktail (AEBSF, Aprotinin, Bestatin, E-64, EDTA, Leupeptin) (Sigma-Aldrich). The protein concentration was measured by bicinchoninic acid analysis (Pierce).

Lysate was pre-cleared by incubation with isotype antibody and bead slurry at 4 °C for 1 h. Affinity purification of MHCII molecules was performed as described previously[62]. In brief, MHCII molecules from cleared lysate were immunoprecipitated by incubation with purified anti-mouse I-A/I-E (20 μg/mg of lysate) (Biolegend) at 4 °C overnight; protein G agarose bead slurry (200 μl) was then added (Invitrogen), with incubation for 5 h at 4 °C. After washing four times with PBS buffer, the MHII molecules were eluted by addition of 1 ml of 10% acetic acid and incubation at room temperature for 4 min with rotation. MHCII–peptide complexes were boiled at 70 °C for 10 min. Peptides and glycopeptides were separated from MHCII by ultrafiltration through a 30 kDa-cutoff membrane filter (Sigma-Aldrich). The eluted MHCII-bound peptides and glycopeptides were subjected to LC–MS/MS analysis in order to assess peptide identity and glycan heterogeneity, as described below.

**Identification of MHCII-bound glycopeptides by MS**. Reduction of samples with dithiothreitol (DTT) (in ammonium bicarbonate buffer at 50 °C for 1 h) and alkylated by iodoacetamide (at room temperature in the dark for 1 h) was followed by digestion with sequence-grade trypsin or chymotrypsin or by no protease treatment. After incubation with or without proteolytic digestion, the samples were further deglycosylated with PNGase F in ¹⁸O water ($H_2^{18}O$) in order to convert the glycan-modified asparagine to an ¹⁸O aspartic acid residue. The sample was further purified by passage through detergent-removal spin columns (Thermo Fisher Scientific).

LC–MS analysis was performed on an Orbitrap Fusion Lumos Tribrid mass spectrometer equipped with an EASY nanospray source and an Ultimate3000 autosampler LC system (Thermo Fisher Scientific). Sample separation was performed on a nano-C18 column (Acclaim pepMap RSLC, 75 μm × 150 mm; C18, 2 μm) via an 80-min gradient of increasing mobile phase B (80% acetonitrile, 0.1% formic acid in distilled $H_2O$) at a flow rate of ∼300 nl/min into the mass spectrometer. For online MS detection, full MS data were first collected at a resolution of 60,000 in Fourier transform (FT) mode, and MS/MS with CID, HCD, or EThcD activation data (all in FT mode) were obtained for each precursor ion by data-dependent scan (top-speed scan, 3 s).

The resulting data were analyzed with Proteome Discoverer 1.4 software (Thermo Fisher Scientific) and the TurboSequest algorithm. For the software search, Sequest parameters were set to allow 10 ppm of precursor-ion mass tolerance and 0.6 Da of fragment-ion tolerance with monoisotopic mass. Digested peptides with up to two missed internal cleavage sites were allowed. For undigested peptides, the minimal and maximal peptide lengths were set at 5 and 30, respectively, with unspecific search (no-enzyme group). Differential modifications of 57.02146, 15.9949, and 2.98826 Da were allowed for alkylated cysteine, oxidated methionines, and ¹⁸O-labeled aspartic acid, respectively. The search was performed against the pSyn gp120 sequence. Any peptide identified from the preliminary software search with a false discovery rate (FDR) > 1% was filtered out. The filtered peptides were further validated manually.

**Glycoproteomic analysis of recombinant glycopeptides**. Glycosylation of the recombinantly expressed glycopeptides was determined by LC–MS/MS analysis. The glycopeptides expressed in 293-F cells were digested with trypsin because of the complexity of the glycosylation, whereas the glycopeptides expressed in GnTI$^{-/-}$ cells were profiled directly. An Orbitrap-Fusion Lumos Tribrid mass spectrometer equipped with an EASY nanospray source and an Ultimate 3000 autosampler LC system was used. LC separation was performed on a nano-C18 column with use of a water/acetonitrile gradient with formic acid. Full MS data were collected at a resolution of 60,000 in FT mode; MS/MS CID, HCD, or EThcD activation data (all in FT mode) were obtained for each precursor ion.

The resulting data were analyzed manually with Byonic software (Protein Metrics). For preliminary data analysis, Byonic parameters were set to allow 20 ppm of precursor-ion mass tolerance and 20 ppm of fragment-ion tolerance with monoisotopic mass. The search was performed against the pSyn gp120 sequence or the target GpepIP sequence with the human/mammalian N-glycan database (the default N-glycan database in Byonic software). Graphic representation of monosaccharide residues is consistent with the symbol nomenclature for glycans (SNFG), which has been broadly adopted by the glycomics community[63].

**I-A$^d$ binding to glycopeptides/peptides in vitro**. In vitro glycopeptide/peptide binding to MHCII molecules was performed as previously described[64]. Purified mouse allele I-A$^d$/CLIP with a 3C protease cleavage site was graciously provided by the NIH tetramer facility. CLIP was removed by treatment of the MHCII monomer with 3C protease (Pierce) for 8 h at room temperature. The cleaved monomer with an empty binding groove was then loaded with the desired peptides or glycopeptides through an exchange reaction. Glycopeptides/peptides (50–300-fold excess over I-A$^d$) were loaded onto I-A$^d$ in citrate buffer (pH 5.0), and the mixture was incubated for 5–6 days at room temperature. At the end of the incubation period, reactions were neutralized with 1 M sodium phosphate buffer (pH 7.5), and the mixture was spun down at maximal speed for 10 min to remove aggregates. Binding was assessed by running samples in pH 3–10 IEF gel (Thermo Scientific).

For western blot, protein complexes were transferred to a PVDF membrane (Bio-Rad) in 0.7% acetic acid buffer for 1 h. After blocking with 3% BSA/PBS, the membrane was incubated with purified anti-mouse I-A/I-E (clone M5/114.15.2) (Biolegend) at 4 °C overnight. Protein bands were visualized by the addition of IRDye secondary antibodies (LI-COR Biosciences), incubation at room temperature for 1 h, and scanning with the Odyssey CLx Imaging System (LI-COR Biosciences). For MS analysis, gel was visualized by Coomassie staining (Bio-Rad). Corresponding bands from IEF gel were excised into smaller pieces, destained, and in-gel tryptic digested. The extracts were purified by passage through a C18-spin column (Nest group) and profiled by LC–MS for peptide/glycopeptide identification as mentioned above.

**RNA-seq analysis**. Mice were immunized three times with GpepIP and pepIP at a 3-week interval. Three weeks after the third immunization, splenic and lymph node cells were isolated and stimulated in vitro with GpepIP and pepIP, respectively, for 3 days. Antigen-responding $CD4^+CD69^+$ T cell populations and $CD4^+CD69^-$ non-responding cells were sorted using the flow cytometry-based cell sorter FACSMelody (BD Biosciences). Total RNA was prepared from these cells using a Quick-RNA Microprep Kit (Zymo Research) according to the manufacturer's protocol. RNA quality was assessed using an Agilent RNA 6000 Nano Kit with an Agilent 2100 Bio analyzer (Agilent Technologies, CA, USA). Library construction and RNA sequencing on BGISEQ-500 platform were conducted at Beijing Genomics Institute (BGI).

The raw sequencing reads were filtered before downstream analyses by removing low-quality reads, adaptor-polluted reads and reads with more than 10% of unknown base. Hierarchical indexing for spliced alignment of transcripts (HISAT)[65] was used to map reads to mm9 reference genome. Gene expression levels were quantified by RSEM V1.2.12[66]. Cluster V3.0 software was used for the hierarchical clustering analysis of the expressed gene. Gene expression cluster was displayed with java Treeview using cluster software[67]. The DEseq2 algorithms[68] were used to detect DEGs between groups. Genes with fold change ≥2 and adjusted $P$-value ≤ 0.05 were considered as significantly differentially expressed. With Gene ontology (GO) annotation, we classified DEGs according to official classification and performed GO functional enrichment using phyper, a function of R[68]. KEGG pathway analysis was also performed by pathway functional enrichment using phyper, a function of R.

**Neutralization assay**. Neutralizing activity of antisera against tier 1 and tier 2 HIV-1 viruses was determined using a luciferase-based TZM-bl cell assay as described elsewhere[29]. Antisera from adjuvant, GpepIP and pepIP primed and three BG505 boosts were tested against Murine leukemia virus (MLV)-pseudo-typed virus, tier 1A MN.3, tier 1B JR-FL and tier 2 BG505/T332N pseudoviruses produced in 293T cells (ATCC). Neutralization titers (50% inhibitory dose, $ID_{50}$) were calculated as the serum dilution at which relative luminescence units (RLUs) were reduced 50% compared to virus control wells (no test sample). MLV-pseudotyped virus was used as negative control for non-HIV-1-specific inhibitory activity in the assay. Monoclonal antibody CH01-31 was used as positive control (shown as antibody concentration).

**BMDC induction and trimer uptake**. BMDCs were generated from bone marrow as described previously[21]. Briefly, bone marrow was flushed out from the tibiae and femurs of 6–8-week-old female BALB/c mice (Taconic Biosciences). Macrophages removed cells were cultured in BMDC induction media (RPMl 1640 supplemented with 10% heat-inactivated FBS, 100 U/ml penicillin, 100 mg/ml stereptomycin, 1% (v/v) MEM non-essential amino acids, 1 mM sodium pyruvate, 2 mM L-glutamine (Thermo Fisher Scientific), 50 µM β-mercaptoethanol (Gibco), and 20 µg/ml GM-CSF (PeproTech) for 8 days. Fresh BMDC induction media were supplemented on days 3 and 6.

For trimer uptake assay, BG505 was first labeled with fluorescein isothiocyanate (FITC) (Sigma-Aldrich) according to the manufacturer's protocol. Briefly, FITC was dissolved in anhydrous dimethyl sulfoxide (DMSO) at a concentration of 1 mg/ml. BG505 was dissolved in 100 mM sodium carbonate buffer (pH 10) at a concentration of 5.4 mg/ml. A 50 molar ratio of FITC to trimer was added and allowed to react at 25 °C for 3 h in dark by agitation. Reaction was quenched by addition of 1 µl of 1 M Tris buffer (pH 7). A desalting procedure using G-25 desalting column (GE Healthcare) was performed following the manufacturer's protocol to further purify fluorophore-labeled BG505. The antibody-mediated immunogen-uptake assay was modified from our previous method[21]. BMDCs were incubated with 20 µg/ml FITC-labeled BG505 with or without pre-incubation with indicated antisera at 37 °C for 2 h. Cells were collected and washed with cold PBS before staining for surface CD11c using anti-mouse CD11c antibody (Biolegend, dilution 1:500, clone N418). Fluorescent signal was detected using flow cytometry.

**Statistical analysis**. All data are presented as mean ± SD values. Student's two-sided $t$-tests for comparison of means was used to compare groups. A $P$ value of <0.05 was considered to be statistically significant.

**Reporting summary**. Further information on research design is available in the Nature Research Reporting Summary linked to this article.

## Data availability

The source data underlying both the main and supplementary figures and uncropped gel images are provided as a Source Data file. The source data underlying Figs. 2a–c and e, 4, Supplementary Table 1 and Supplementary Figs. 2, 4 and 5 are provided as a Source Data file. All the mass spectrometry raw data generated for this study and Supplementary Data 1–4 have been deposited in Mendeley Data repository with the identifier DOI: 10.17632/b8sk9zc7xj.2. All other data generated during the current study are available from the corresponding author upon requests.

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

## Acknowledgements

We thank K. Moremen for the PNGase F enzyme. We thank B. Haynes and K. Campbell for early discussions of the project and J. McCoy for editorial assistance. Purified mouse allele I-A^d/CLIP with a 3C protease cleavage site was provided by the NIH tetramer core facility (Emory University). Funding was provided by National Institutes of Health grants R01AI123383 and P41GM103490.

## Author contributions

L.S. and F.Y.A. conceived the study. L.S., C.C.L., M.T., and F.Y.A. designed the experiments; L.S., D.R.M., M.I., A.V.P., J.A., A.O., P.L.W., and J.A.D. performed the research; L.S., M.I., J.A., M.T., and F.Y.A. analyzed the data; and L.S. and F.Y.A. wrote the paper.

## Competing interests

The authors declare no competing interests.
