## [Peer Review File · Nature Communications]

Reviewers' comments:

Reviewer #1 (Remarks to the Author):

In this paper, Sun et al investigate CD4+ T cell recognition of glycopeptides derived from the HIV envelope protein gp120 (a protein which is decorated with numerous glycans). Although much work has been done over the years to characterise antibody recognition of HIV Env glycans, very little has been done in terms of T cell recognition of glycans in the context of HIV immunity and how glycan-specific T cells could be harnessed to enhance neutralising antibody responses. Thus this study will be of great interest to both the HIV antibody field as well as the broader vaccinology field. I found this paper to be very interesting and worth publishing - however, it is currently missing some key data to support the author's conclusions and provide more mechanistic insight into how glycoconjugate immunisation can enhance Env antibody responses.

The main claims of the study are that glycans present on gp120 glycopeptides presented by MHC-II are specifically recognised by CD4+ T cells (whether the T cells recognise glycan only or glycan+peptide, or indeed whether the stimulated cells were MHC-II-restricted did not come across clearly to me), and that gp120 glycopeptide immunisation modulates helper T cell differentiation which in turn results in more effective help for humoral anti-gp120 responses in vivo. I am persuaded that GpepIP (as opposed to naked peptide) immunisation results in superior Env-specific antibody responses, however definitive evidence for this effect being CD4+ T cell-mediated is lacking.

Overall, the study is of high technical quality and the methods section provides sufficient detail for replication. The question being addressed and the results are generally presented clearly in an understandable way with some exceptions (see comments below).

Specific comments:

1. I do not have direct experience of glycopeptide identification by mass spectrometry. Thus to the best of my knowledge the experiments performed for Fig2 and the methods are of high quality. However, the link from this section to the next section (Fig3) is not clear, as there is no data definitively showing that the elicited GpepIP-specific CD4+ T cells are MHC-II-restricted. There are examples in the literature of MHC-I-restricted CD4+ T cells in mice. There is also an example of T cells recognising glycan alone in the absence of MHC but requiring APCs (Muthukkumar et al 2003 Vaccine). Thus it is important to show that the GpepIP-induced cells are MHC-II-restricted. This could be done using anti-class II blocking Abs during stimulation, and also by using GpepIP-pulsed MHC-II-/- cells as APCs to rule out 'presentation' by alternative cell-surface proteins.

I am also unsure whether the GpepIP-specific responses shown in Fig3 required APCs? Were the stimulations here done in the presence of mito-C-treated APCs? If not, is the observed response due to presentation of GpepIP on CD4+ T-expressed MHC-II or direct recognition of glycoconjugates in solution?

Another query is whether the GpepIP-specific CD4+ T cells also recognise peptides containing a different amino acid sequence but containing similar glycan structures? Have alternative peptides

with the same glycan structure been tested? I note that the BG505 sequence is very different from the GpepIP sequence – does the equivalent peptide from BG505 also bind to MHC-II and stimulate the same GPepIP-specific T cells? Is the glycan on the equivalent BG505 peptide the same as the one on GpepIP?

In the case that T cells recognise glycan alone they could potentially be activated by FDCs presenting whole antigen. The GpepIP-specific cells could also potentially recognise native Env protein and enhance infection.

In the case that glycan is recognised in the context of MHC-II (no involvement of peptide), this would potentially have implications for self-recognition and tolerance control.

2. The data shown in Figure 4 indicate that GpepIP induces stronger Th2 and Th17 enrichment than pepIP. Is the glycan found on GpepIP also found on gut microbes? If so, could the authors discuss the possibility that GpepIP induces an expansion of a pre-existing microbe glycan-specific T cells (which may be enriched in Th17 populations). It has been shown that HIV-specific memory CD4+ cells primed by microbes are present in unexposed individuals (Su et al 2013 Immunity).

3. The data shown in Figure 6 and 7 purport to show that GPepIP-specific CD4+ T cells provide better help for Env-specific antibody production/function. It is evident that GpepIP immunisation achieves this, but not that CD4+ T cells are the main actors in the effect. An experiment where animals are primed with GpepIP OR pepIP OR adjuvant, then CD4+ T cells are isolated and adoptively transferred into naïve mice which are then 'boosted' with BG505 would more definitively show whether the pre-existing GpepIP-specific cells provide superior help for Ab production and function.

Minor comments:

1. The first introductory sentence "Acquired immunodeficiency syndrome (AIDS) caused by human immunodeficiency virus-1 (HIV-1) remains the most common cause of death from an infectious agent" is not factually correct. As I understand, tuberculosis is currently the most common infectious cause of death (having overtaken HIV/AIDS in recent years). The authors could reframe this sentence to emphasise number of people living with HIV, % without treatment or number of deaths per year, citing WHO/UNAID latest figures.

2. The reference (5) in the sentence starting "HIV Env trimer is highly glycosylated..." does not support the sentence's conclusions. This reference may have been meant for the previous sentence? Leonard et al JBC 1990 may be more suitable here.

3. In the text, GpepIP is specifically designated to the sequence gp120-KLDVVPIDNNN187TSY. However, in supplementary figure S5 a sequence is shown with c-terminal ...TSYR. Similarly, the sequences of peptides excised from IEF gels shown in Fig2E have C-term R. It is unclear what the exact sequence definition of GPepIP is – can this be clarified?

4. Fig 2D may work better in black and white for contrasting the different bands.

5. On p11, the concluding sentence states the results show that GPepIP modulates CD4+ T cell differentiation. This is jumping the gun a bit as the results in that section do not show this (the results in the subsequent section do though).

6. On p16, the sentence "After a 5-day GpepIP or PepIP antigen stimulation of T cells from GpepIP or PepIP immunised mice, supernatants were harvested for a multiplex-based cytokine measurement." is unclear. Were cells from GpepIP-immunised mice stimulated with GpepIP OR pepIP? Or just GPepIP (this is what I presume but it is not clear in the text or in the figure 5 legend – 'respectively' could be added to the sentence to clarify this)?

7. p17, the sentence 'As the cytokines were detected...' would read better as "When cytokines were detected..." to more clearly reflect the meaning.
8. p19, the sentence "Compared to adjuvant and pepIP, GpepIP primary immunization induced superior germinal center (GC) response, defined by a significantly increased percentage of GL7+Fas+ B cells (Fig. 6D)". Is this the percentage within total B cells? This could at least be clarified in the Fig 6 legend or on the y-axis of Fig6D.
9. The colour system in Fig6 changes from panel to panel (i.e. B and D are similar. Then C and E-G use different colours) – it would best to harmonise the colour system.
10. P25 – "Therefore, gp120 glycan shield will significantly skew CD4+ T cell epitope "hot spots". Thus, it is plausible to evaluate the importance of gp120 glycopeptide specific CD4+ T cells." I don't fully understand this – did the authors mean to say 'reasonable' instead of 'plausible'?

Reviewer #2 (Remarks to the Author):

This manuscript defines the role of a glycopeptide (GpepIP) derived from the HIV-1 gp120 glycoprotein. The identified peptide binds MHCII and stimulates specific T cell proliferation. I find the manuscript is interesting since it shows that a glycosylated Env-derived peptide can function as a T helper epitope for the immune response. It was furthermore interesting to see that the T cell differentiation pattern of GpepIP- and pepIP-immunized mice after in vitro stimulation with respective peptide were different (ie Th17-differentiation was more prominent in GpepIP stimulated T cells). However, in order to directly compare a glycosylated peptide with a non-glycosylated peptide for their role on B cell activation (fig 6 and 7), a naturally non-glycosylated peptide should be used as a prime since the non-glycosylated peptide used here (pepIP) might not be present in the mice after processing of the gp120 antigen. If this would be the case, the mice that are primed with pepIP will not get a t-cell prime that is shared with the protein used for boost.

Specific comments:

1. In the introduction, the following sentence should be modified since it can be misunderstood that glycans are the only target of bNAbs:
"An increasing number of studies have illustrated that in addition to providing a shield to avoid immune responses, gp120 glycans are the major "sites of vulnerability" targeted by broadly neutralizing antibodies (bNAbs)..."
2. It should be clarified if the results in Figure 1D are from the same experiment described in 1A and 1B. I would also suggest to make Figure 1D to Figure 1C so the T cell response is discussed together, before the serum response is shown.
3. For the serum results in figure 1, in general, I am confused with what the authors want to show and what the conclusions are, this should be explained better in the results section. Although low (with OD values of <1) I am surprised to see antibody responses elicited by the gp120 peptide pool that cross react to the full protein since I would expect that the peptides used for immunization do not have the same 3D structure as they have when part of the protein. Why study these cross-

reactive serum responses? Is it to show that Ab-responses can develop against glycans? Is it not better then to compare serum responses to the glycopeptide pool used for immunization and compare to the same peptide pool wo glycans? The rational behind the Ab analysis is not explained. It would also be valuable if the authors could specify the strain of the gp120 used for generating the pool and the intact gp120 used for ELISA so it is clear if they are the same or different.

4. In Figure 1C it is indicated that all cross-reactive ab-responses are specific against the glycosylated protein since this is the only ag that can inhibit binding but then in Figure 1G and H you show binding of IgG2a and IgG3 to DG gp120. How is that explained? It would be interesting to see an ELISA like in Figure 1C but against DG gp120, with gp120 and DG gp120 to inhibit binding.

5. What is the dilution of the serum in the ELISA of Figure 1C?

6. It is described that a variety of peptides were identified in Table S1, why is the KLDVVPIDNNN187TYS selected to be used in the rest of the paper and not one of the others?

7. Regarding Figure 3 I have the same comment as for Figure 1, explain why you examine Ab-responses elicited by the peptide that cross react with the full antigen? If you want to study the isotpye of the response, why not do ELISA against the peptide itself so the titers are higher, as in Fig 3E and 3F? The titers to the full protein are very low and it is hard to see what is above background in Figure 3 H, I and J. If figure H, I and J is kept, a control for background should be included.

8. In the experiment described in Figure 6, it is indicated in the figure that the peptide is derived from Clade B and then you use BG505 trimers (Clade A) for boost. Is the peptide sequence used for prime conserved between the strains?

9. Figure 6B: It is mentioned in the result section that the GpepIP-response elicited in the adjuvant group alone is potentially due to the BG505 Env boost. I think it is very important that this is evaluated by performing a control experiment. Otherwise it is possible that the enhanced responses in Fig 6 D – G is due to some unspecific activation of cells.

10. Figure 6C (and Figure 7B). Is the ELISA against the BG505 Env trimer or BG505 peptide? The title of the graph just indicates the strain name of the virus.

11. Figure 6C. Again I am surprised that there are Ab-responses against the BG505 Env trimer(?) before the trimer is given (day 28). I think this should be discussed more, especially since a peptide from a different strain (clade B) has been used as prime.

12. In Figure 6D it is unclear what is being measured. In the results section it is noted that this is Env trimer-specific B cell responses, but in the figure it says GC B cells - total GC B cells? It is likely that the peptide has activated B cells specific only to the peptide and not to the Env trimer and/or there can be un-specific responses going on at the same time so the B cells should be stained by flow cytometry using a biotinylated Env trimer in order to confirm they are trimer Env-specific.

13. The last sentence in the discussion claims that GpepIP induces T cells that activate B cells with

protective antibodies. I don't think the word protective is suitable since the MN.3 virus is the lowest bar of neutralization (T cell-line-adapted virus, Clade 1A) and does not reflect natural neutralization of primary viruses.

Specific minor comments:

- "HIV" should be changed to "HIV-1" throughout the paper.
- In some figure legends the result of the experiment is included and should be removed.
- How many times is each experiment repeated? This information should be added to fig legends.
- The colors for pepIP and GpepIP are red and blue respectively in figure 6B and D but it the opposite in figure 6E-G. It would be more comprehensive if they are the same.
- The titles of the y-axes should be looked over. Fig 6D indicates for example "% of GL7+Fas+ cells" and should probably be "% GL7+Fas+ of total B cells" or "% Env-specific B cells of GC B cells".

Response to reviewers' comments:

We thank the reviewers for thoroughly reading our manuscript and providing insightful comments that have contributed to this much improved report. We have attempted to address every comment in our revised manuscript and in our responses below.

Reviewer

#1

In this paper, Sun et al investigate CD4+ T cell recognition of glycopeptides derived from the HIV envelope protein gp120 (a protein which is decorated with numerous glycans). Although much work has been done over the years to characterise antibody recognition of HIV Env glycans, very little has been done in terms of T cell recognition of glycans in the context of HIV immunity and how glycan-specific T cells could be harnessed to enhance neutralising antibody responses. Thus this study will be of great interest to both the HIV antibody field as well as the broader vaccinology field. I found this paper to be very interesting and worth publishing - however, it is currently missing some key data to support the author's conclusions and provide more mechanistic insight into how glycoconjugate immunisation can enhance Env antibody responses.

The main claims of the study are that glycans present on gp120 glycopeptides presented by MHC-II are specifically recognised by CD4+ T cells (whether the T cells recognise glycan only or glycan+peptide, or indeed whether the stimulated cells were MHC-II-restricted did not come across clearly to me), and that gp120 glycopeptide immunisation modulates helper T cell differentiation which in turn results in more effective help for humoral anti-gp120 responses in vivo. I am persuaded that GpepIP (as opposed to naked peptide) immunisation results in superior Env-specific antibody responses, however definitive evidence for this effect being CD4+ T cell-mediated is lacking.

Overall, the study is of high technical quality and the methods section provides sufficient detail for replication. The question being addressed and the results are generally presented clearly in an understandable way with some exceptions (see comments below).

Comment 1:

I do not have direct experience of glycopeptide identification by mass spectrometry. Thus to the best of my knowledge the experiments performed for Fig2 and the methods are of high quality. However, the link from this section to the next section (Fig3) is not clear, as there is no data definitively showing that the elicited GpepIP-specific CD4+ T cells are MHC-II-restricted. There are examples in the literature of MHC-I-restricted CD4+ T cells in mice. There is also an example of T cells recognising glycan alone in the absence of MHC but requiring APCs (Muthukkumar et al 2003 Vaccine). Thus it is

important to show that the GpepIP-induced cells are MHC-II-restricted. This could be done using anti-class II blocking Abs during stimulation, and also by using GpepIP-pulsed MHC-II-/- cells as APCs to rule out 'presentation' by alternative cell-surface proteins.

Response:

We thank the reviewer for the critical point regarding whether the GpepIP-induced CD4 T cell response is MHC-II-restricted. We followed the reviewer's suggestion using anti-class II blocking Abs during CD4 T cell response to GpepIP and pepIP. We have performed the experiment and the results clearly showed that MHCII blocking significantly inhibited CD4+ T cells response to GpepIP and pepIP in a dose-dependent manner, suggesting that GpepIP/pepIP-specific CD4+ T cells are MHC-II-restricted. We have put these results in supplemental figure 6a and b. If deemed more appropriate, we could certainly move this figure into main text.

Comment 2:

I am also unsure whether the GpepIP-specific responses shown in Fig3 required APCs? Were the stimulations here done in the presence of mito-C-treated APCs? If not, is the observed response due to presentation of GpepIP on CD4+ T-expressed MHC-II or direct recognition of glycoconjugates in solution?

Response:

The GpepIP-specific T cell response in Fig 3 was done in the presence of mitomycin-C treated APCs. We have now added the description "in the presence of mitomycin C-treated APCs" in the figure legend. In addition, we have now supplemented this data showing that GpepIP/pepIP-specific CD4+ T cells are MHC-II-restricted.

Comment 3:

The data shown in Figure 6 and 7 purport to show that GPepIP-specific CD4+ T cells provide better help for Env-specific antibody production/function. It is evident that GpepIP immunisation achieves this, but not that CD4+ T cells are the main actors in the effect. An experiment where animals are primed with GpepIP OR pepIP OR adjuvant, then CD4+ T cells are isolated and adoptively transferred into naïve mice which are then 'boosted' with BG505 would more definitively show whether the pre-existing GpepIP-specific cells provide superior help for Ab production and function.

Response:

We thank the reviewer for suggesting this experiment. Despite the instances that peptide priming followed by antigen boosting are used to evaluate CD4 T cell help (Sette et al, 2008 *Immunity* and Alam et al, 2014 *J virol*), we agree with the reviewer that more experiments are needed to definitively show whether the pre-existing GpepIP-specific cells provide superior help for Ab production and function. To directly reveal the role of GpepIP specific CD4+ T cells in helping HIV Env trimer specific humoral immune responses, we performed both CD4+ T cell adoptive transfer

and CD4+ T cell depletion experiments. As the reviewer suggested, CD4+ T cells were purified from donor mice immunized with GpepIP, pepIP or adjuvant and adoptively transferred into naïve recipient mice followed by BG505 trimer immunization. The results showed that Env trimer-specific IgG titer was significantly higher in mice that received CD4+ T cells from GpepIP-immunized donor mice than those that received CD4+ T cells from pepIP- and adjuvant-immunized mice. In addition, mice were immunized with GpepIP, pepIP or adjuvant followed by BG505 trimer boost. Depletion of CD4+ T cells during BG505 trimer immunization by injecting anti-CD4 mAb completely diminished the helping effect of GpepIP priming. All these data strongly demonstrate that GpepIP epitope helping HIV Env trimer specific humoral immune responses is CD4+ T cells dependent. We have put these results in supplemental figure 9a-d. If deemed appropriate, we could move these important data into main text.

Comment 4:

Another query is whether the GpepIP-specific CD4+ T cells also recognise peptides containing a different amino acid sequence but containing similar glycan structures? Have alternative peptides with the same glycan structure been tested?

Response:

We believe that this important point aligns with understanding structural and molecular requirements for the mechanisms laid out here and thus deserves a separate manuscript. In this study, we identified an MHCII-presented glycopeptide epitope (GpepIP) eliciting glycopeptide-specific CD4+ T cell response. Our data (Figure 3) demonstrate that CD4+ T cells recognition of GpepIP is glycan dependent, evidenced by: 1) loss of glycans impairs the T cell response to the glycopeptide epitope; 2) alteration of glycan structure dampens the T cell response; 3) addition of glycans to the peptide blocks the T cell response. Since we haven't tested alternative peptides with the same glycan structure or obtained monoclonal T cells, we cautiously stated in the manuscript that GpepIP-specific CD4+ T cells are glycopeptide-specific or glycan-dependent.

We also added a statement in the discussion: "Future studies are required to reveal the recognition specificities of GpepIP specific CD4+ T cells, such as evaluating alternative peptides containing the same glycan structure."

Finally, as we discuss in our response to comment 5 below (and further discussions as part of second reviewers comments), since BG505 does not share the peptide sequence of GpepIP and yet induce booster response, we postulate that the peptide portion of the glycopeptide epitope is required for MHCII binding and presentation, however, it is not required for TCR recognition. In this manuscript we stop just short of making this statement as it requires a series of experimentation that we thought would make this manuscript unwieldy.

Comment 5:

I note that the BG505 sequence is very different from the GpepIP sequence – does the equivalent peptide from BG505 also bind to MHC-II and stimulate the same GPeP-

specific T cells? Is the glycan on the equivalent BG505 peptide the same as the one on GpepIP?

Response:

As the reviewer points out, BG505 equivalent sequence is different from the GpepIP sequence. Moreover, the glycosylation site and glycan structures are different between GpepIP and BG505 equivalent sequence as well. Thus, further assessments are required to reveal whether the equivalent peptide from BG505 also binds to MHCII and stimulates the same GpepIP-specific T cells.

We have added sentences and citations in the discussion, as “The GpepIP region is highly variable across gp120s from different clades. The equivalent peptide sequence and glycans from BG505 are significantly different from GpepIP. We have shown that GpepIP specific CD4+ T cells cross-react with shared glycopeptide and/or glycan epitopes generated after BG505 processing in APCs. However, it is still important to investigate whether the equivalent glycopeptide from BG505 also binds to MHCII and stimulates the GpepIP-induced CD4+ T cells.”

Comment 6:

In the case that T cells recognise glycan alone they could potentially be activated by FDCs presenting whole antigen. The GpepIP-specific cells could also potentially recognise native Env protein and enhance infection.

Response:

The recognition of GpepIP by T cells is MHCII processing/presentation dependent as we *now* show that MHCII blocking inhibits CD4+ T cells response to GpepIP (Fig S6). Moreover, the GpepIP epitope is an MHCII-bound glycopeptide identified by MHCII immunoprecipitation and binds to MHCII molecules *in vitro* (Figure 2). Thus, GpepIP-specific cells do not recognize Env protein in its native form and enhance infection.

Comment 7:

In the case that glycan is recognised in the context of MHC-II (no involvement of peptide), this would potentially have implications for self-recognition and tolerance control.

Response:

We think this a very important point and we addressed more on this in the discussion, as “While the glycan portion of GpepIP is a product of host glycosylation machinery, it is still antigenic. Self-recognizing T cells are not expected to be a concern due to the fact that the dominant oligo/high-mannose structures on HIV-1 Env or GpepIP are markedly distinct from predominant host (mammalian) N-glycans (Moremen et al, 2012, *Molecular cell biology*)”. In addition, similar to bNAbs targeting glycans, while the potential risk of inducing autoreactive antibody production is a concern, most glycan-targeting bNAbs isolated from HIV infected individuals are not autoreactive, and “very few glycan-

targeting bNAbs are autoreactive but still protective (Haynes et al, 2010, *Nat Struct Mol Biol*). We plan to address whether GpepIP specific CD4+ T cells have self-recognition or any potential autoreactive T cells in the next phase of our study.

Comment 8:

The data shown in Figure 4 indicate that GpepIP induces stronger Th2 and Th17 enrichment than pepIP. Is the glycan found on GpepIP also found on gut microbes? If so, could the authors discuss the possibility that GpepIP induces an expansion of a pre-existing microbe glycan-specific T cells (which may be enriched in Th17 populations). It has been shown that HIV-specific memory CD4+ cells primed by microbes are present in unexposed individuals (Su et al 2013 Immunity).

Response:

This is a very interesting point. We are aware of commensal microbiota play essential roles in shaping immune system and T cell development. There are also studies reported that microbiota glycans induce T cell differentiation, such as that polysaccharide A (PSA) derived from the human commensal *Bacteroides fragilis* promotes human Treg function (Round et al, 2010, *Proc Natl Acad Sci U S A* and Telesford et al, 2015, *Gut Microbes*). However, the N-glycan composition and structures in prokaryotic cells are significantly different from that in eukaryotic cell (Essentials of Glycobiology; Latousakis et al, 2018, *Int J Mol Sci* and Geva-Zatorsky et al, 2015, *Nat Med*). Therefore, to our best knowledge, glycans on GpepIP (Man5GlcNAc2) are not present in gut microbes.

Comment 9:

The first introductory sentence “Acquired immunodeficiency syndrome (AIDS) caused by human immunodeficiency virus-1 (HIV-1) remains the most common cause of death from an infectious agent” is not factually correct. As I understand, tuberculosis is currently the most common infectious cause of death (having overtaken HIV/AIDS in recent years). The authors could reframe this sentence to emphasise number of people living with HIV, % without treatment or number of deaths per year, citing WHO/UNAID latest figures.

Response:

We thank the reviewer for point this out. We have made changes in the introduction as “Acquired immunodeficiency syndrome (AIDS) caused by human immunodeficiency virus-1 (HIV-1) remains one of the leading causes of death from an infectious agent. Globally, 37.9 million people were living with HIV by the end of 2018 and HIV infection has contributed more than 35 million deaths since its emergence”.

Comment 10:

The reference (5) in the sentence starting “HIV Env trimer is highly glycosylated...” does not support the sentences conclusions. This reference may have been meant for the previous sentence? Leonard et al JBC 1990 may be more suitable here.

Response:

Thank you for pointing out this error. We have now cited the relevant reference.

Comment 11:

In the text, GpepIP is specifically designated to the sequence gp120-KLDVVPIDNNN187TSY. However, in supplementary figure 5, a sequence is shown with c-terminal ...TSYR. Similarly, the sequences of peptides excised from IEF gels shown in Fig2e have C-term R. It is unclear what the exact sequence definition of GPepIP is – can this be clarified?

Response:

To clarify: We used sequence KLDVVPIDNNNTSY in the text because it is identified from MHCII immunoprecipitation with no enzyme treatment (supplemental Table 1), which is the naturally processed epitope. However, since an “R” residue was presented in enzyme treated group, to be more inclusive, we did express the sequence containing the “R” at the end. We now indicate in the main text that our designated GPepIP is “KLDVVPIDNNNTSYR”.

Comment 12:

Fig 2D may work better in black and white for contrasting the different bands.

Response:

We have changed Fig 2d into black and white.

Comment 13:

On p11, the concluding sentence states the results show that GPepIP modulates CD4+ T cell differentiation. This is jumping the gun a bit as the results in that section do not show this (the results in the subsequent section do though).

Response:

We agree with the reviewer and we have removed the concluding sentence regarding CD4+ T cell differentiation.

Comment 14:

On p16, the sentence “After a 5-day GpepIP or PepIP antigen stimulation of T cells from GPepIP or PepIP immunised mice, supernatants were harvested for a multiplex-based

cytokine measurement.” is unclear. Were cells from GpepIP-immunised mice stimulated with GpepIP OR pepIP? Or just GPepIP (this is what I presume but it is not clear in the text or in the figure 5 legend – ‘respectively’ could be added to the sentence to clarify this)?

Response:

We have followed the reviewer’s advice and added “respectively” in both the main text and figure 5 legend to avoid the ambiguity.

Comment 15:

p17, the sentence ‘As the cytokines were detected...’ would read better as “When cytokines were detected...” to more clearly reflect the meaning.

Response:

Thanks for the suggestion. We have changed it to “When cytokines...”.

Comment 16:

p19, the sentence “Compared to adjuvant and pepIP, GpepIP primary immunization induced superior germinal center (GC) response, defined by a significantly increased percentage of GL7+Fas+ B cells (Fig. 6D)”. Is this the percentage within total B cells? This could at least be clarified in the Fig 6 legend or on the y-axis of Fig6D.

Response:

In Fig 6d, we analyzed the GL7+Fas+ (GC) B cells within total B cells. We have changed the figure 6 legend as “the percentage of GL7⁺Fas⁺ B cells among B220⁺ B cells” and also changed the y-axis of Fig 6d as “% GL7⁺Fas⁺ of total B cells”.

Comment 17:

The colour system in Fig6 changes from panel to panel (i.e. B and D are similar. Then C and E-G use different colours) – it would best to harmonise the colour system.

Response:

We have changed the colors in this figure to be more consistent between groups.

Comment 18:

P25 – “Therefore, gp120 glycan shield will significantly skew CD4+ T cell epitope “hot spots”. Thus, it is plausible to evaluate the importance of gp120 glycopeptide specific CD4+ T cells.” I don’t fully understand this – did the authors mean to say ‘reasonable’

instead

of

'plausible'?

Response:

We have changed this to “reasonable” for clarity as suggested by the reviewer.

Reviewer #2

This manuscript defines the role of a glycopeptide (GpepIP) derived from the HIV-1 gp120 glycoprotein. The identified peptide binds MHCII and stimulates specific T cell proliferation. I find the manuscript is interesting since it shows that a glycosylated Env-derived peptide can function as a T helper epitope for the immune response. It was furthermore interesting to see that the T cell differentiation pattern of GpepIP- and pepIP-immunized mice after in vitro stimulation with respective peptide were different (ie Th17-differentiation was more prominent in GpepIP stimulated T cells).

Comment 1:

In the introduction, the following sentence should be modified since it can be misunderstood that glycans are the only target of bNAbs: “An increasing number of studies have illustrated that in addition to providing a shield to avoid immune responses, gp120 glycans are the major “sites of vulnerability” targeted by broadly neutralizing antibodies (bNAbs)...”

Response:

As advised, we have now changed the sentence to “gp120 glycans can be the major “sites of vulnerability” targeted by broadly neutralizing antibodies (bNAbs)”.

Comment 2:

It should be clarified if the results in Figure 1D are from the same experiment described in 1A and 1B. I would also suggest to make Figure 1D to Figure 1C so the T cell response is discussed together, before the serum response is shown.

Response:

As recommended, we have made Figure 1d, e to Figure 1c, d to better separate T cell cytokines and serum response results. This will also help to clarify that results from Figure 1d, e are from the same experiment with Figure 1a, b.

Comment 3:

For the serum results in figure 1, in general, I am confused with what the authors want to show and what the conclusions are, this should be explained better in the results

section. Although low (with OD values of <1) I am surprised to see antibody responses elicited by the gp120 peptide pool that cross react to the full protein since I would expect that the peptides used for immunization do not have the same 3D structure as they have when part of the protein. Why study these cross-reactive serum responses? Is it to show that Ab-responses can develop against glycans? Is it not better then to compare serum responses to the glycopeptide pool used for immunization and compare to the same peptide pool wo glycans? The rational behind the Ab analysis is not explained. It would also be valuable if the authors could specify the strain of the gp120 used for generating the pool and the intact gp120 used for ELISA so it is clear if they are the same of different.

Response:

Thank you for these insightful points. In the current study, gp120 glycopeptides pool were generated by protease digestion and used in the immunization. Since gp120 proteins are heavily glycosylated with various numbers (from 23 to 26) of N-linked glycosylation sites across the protein backbone, the glycopeptides pool covers most of the protein sequence. Additionally, those glycan epitopes targeted by B cells are shared between glycopeptides and gp120 protein. Therefore, the antibody responses elicited by the gp120 glycopeptides pool could cross react to the full protein.

The reasons we evaluate cross-reactive serum responses to full protein are 1) as reviewer pointed out to show that Ab-responses can develop against glycans; 2) to study the immunogenicity of our immunogens in eliciting antibodies against HIV-1 natural antigens gp120 or Env trimer proteins for future advanced vaccine design.

We have added sentences to better explain the rational and conclusion of this Ab analysis as “We next investigated if enriched glycopeptide-specific CD4⁺ T cell response induces glycan/glycopeptide-dependent antibody response against HIV-1 envelope antigen gp120.

These results indicate that substantial glycan-dependent antibodies were elicited after glycopeptide pool immunization.”

We have also specified the strain of the gp120 (JR-FL, Clade B) used for both generating the glycopeptide pool and for ELISA.

Comment 4:

In Figure 1C it is indicated that all cross-reactive ab-responses are specific against the glycosylated protein since this is the only ag that can inhibit binding but then in Figure 1G and H you show binding of IgG2a and IgG3 to DG gp120. How is that explained? It would be interesting to see an ELISA like in Figure 1C but against DG gp120, with gp120 and DG gp120 to inhibit binding.

Response:

In Figure 1c of the initial submission, DG-gp120 also showed inhibitory effect. The inhibition was subtle, which may have been due to a higher concentration of serum

used (1:800). Therefore, in this revised manuscript, we have updated the inhibition ELISA result (now Figure 1e) using a lower serum concentration (1:1600). In the updated figure, it is more evident that both gp120 and DG-gp120 inhibited binding with gp120 having greater inhibition.

In addition, as the reviewer suggested, we also performed the inhibition ELISA against DG-gp120 in the supplemental Figure 1c. “However, the inhibition between intact gp120 and DG-gp120 against binding to DG-gp120 was comparable (supplemental Fig. 1c).”

Comment 5:

What is the dilution of the serum in the ELISA of Figure 1C?

Response:

The serum dilution used in the previous Figure 1c is 1:800. But we updated this figure (now Figure 1e) using a serum dilution of 1:1600. We have specified it in the figure legend together with all other serum dilution in Figure 3g, 6c, 7b and 7c.

Comment 6:

It is described that a variety of peptides were identified in Table S1, why is the KLDVVPIDNNN187TYS selected to be used in the rest of the paper and not one of the others?

Response:

Glycopeptide KLDVVPIDNNNTSYR stood out because it was identified in all protease-treated and untreated pools of MHCII bound epitopes (Table S1). More importantly, being identified from no enzyme group, it reflects what is naturally presented by MHCII. We have added this justification in the manuscript, as “**One particular** glycopeptide epitope... was identified in **both** protease-treated and untreated pools.... This glycopeptide identified from protease-untreated sample represents a naturally processed T cell epitope.”

Comment 7:

Regarding Figure 3 I have the same comment as for Figure 1, explain why you examine Ab-responses elicited by the peptide that cross react with the full antigen? If you want to study the isotpye of the response, why not do ELISA against the peptide itself so the titers are higher, as in Fig 3E and 3F? The titers to the full protein are very low and it is hard to see what is above background in Figure 3 H, I and J. If figure H, I and J is kept, a control for background should be included.

Response:

As also noted above, the reason we evaluate cross-reactive serum responses to full protein is to study the immunogenicity of our immunogens in eliciting antibodies against natural HIV-1 antigens gp120 or Env trimer proteins for future advanced vaccine design. To better convey this rationale, we have added a sentence to explain, as “...,”

suggesting that GpepIP is more immunogenic than pepIP in eliciting gp120 specific antibody response.”

Based on the same rationale, we assessed the IgG subclasses of GpepIP-immunized antisera against full antigens. To avoid non-specificity, we performed the experiments in Figure 3h, i and j by adding control groups as background using serum from naïve mice against gp120 and DG-gp120. For clear presentation, we now show gp120 and DG-gp120 groups in the figures after subtracting background signal. We described the method in the figure legend.

As reviewer suggested, we also performed the experiment against the glycopeptide and peptide. The results demonstrated that GpepIP-immunized sera showed much greater binding to GpepIP than pepIP in all three IgG subclasses. This is expected because our results in Figure 3e and 3f showed that GpepIP-immunized antisera had remarkably higher total IgG against GpepIP than pepIP. Those data have been added as supplemental Figure 6c-e.

Comment 8:

In order to directly compare a glycosylated peptide with a non-glycosylated peptide for their role on B cell activation (fig 6 and 7), a naturally non-glycosylated peptide should be used as a prime since the non-glycosylated peptide used here (pepIP) might not be present in the mice after processing of the gp120 antigen. If this would be the case, the mice that are primed with pepIP will not get a t-cell prime that is shared with the protein used for boost.

Response:

We thank the reviewer for this important point. We agree that evaluating a naturally processed non-glycosylated peptide is important. Thus we have now added the statement in the discussion “It is important to compare GpepIP epitope with a naturally processed non-glycosylated peptide in helping Env antibody response as part of future studies.”

For the scope of this manuscript, we believe using pepIP is justifiable. First, pepIP contains the same peptide sequence of GpepIP with only difference in glycans, which makes a better comparison between GpepIP and pepIP than GpepIP with other peptides different in both amino acid sequence and glycans. Including a new peptide would require analyzing it for its MHCII binding and comparing with GpepIP in a very indirect fashion for their MHCII allelic preference/specificity as well as binding affinity. In addition, we have shown that pepIP bound to MHCII (Fig2c) and pepIP immunization stimulated T cell response (Fig 3d and 6b) specific to pepIP but not GpepIP. Since the naked peptide is a T cell epitope, we could directly compare glycopeptide- and peptide-specific helper T cells in helping trimer antibody response. All said, we still think that the reviewer’s suggestion addresses a very important point, albeit requiring a new set of experimental design and prerequisite experiments to avoid false negative/positive results.

Comment 9:

In the experiment described in Figure 6, it is indicated in the figure that the peptide is derived from Clade B and then you use BG505 trimers (Clade A) for boost. Is the peptide sequence used for prime conserved between the strains?

Response:

The equivalent peptide sequence from BG505 trimer is different from the GpepIP sequence. The reason we use BG505 trimer for boosting is that it is a soluble, native-like, and well-ordered Env trimer. It is structurally and antigenically well characterized and widely used for elicitation of bNAbs response. Therefore, we used BG505 trimer to evaluate the potency of glycopeptide epitope specific CD4 T cells in helping Env trimer humoral immune responses and functional antibody production. Furthermore, by using a heterologous trimer, our results demonstrated that GpepIP epitope shows greater and broader potency in helping cross-clade Env trimer antibody responses than pepIP epitope.

We have also added sentences in the discussion, as “The GpepIP region is highly variable across gp120s from different clades. The equivalent peptide sequence and glycans from BG505 are significantly different from GpepIP. We have shown that GpepIP specific CD4+ T cells cross-react with shared glycopeptide and/or glycan epitopes generated after BG505 processing in APCs. However, it is still important to investigate whether the equivalent glycopeptide from BG505 also binds to MHCII and stimulates the GpepIP-induced CD4+ T cells.”

Comment 10:

Figure 6B: It is mentioned in the result section that the GpepIP-response elicited in the adjuvant group alone is potentially due to the BG505 Env boost. I think it is very important that this is evaluated by performing a control experiment. Otherwise it is possible that the enhanced responses in Fig 6 D – G is due to some unspecific activation of cells.

Response:

In the revised manuscript we now demonstrate that GpepIP stimulates CD4 T cell response dependent on MHCII presentation (MHCII blocking experiment) (supplemental Figure 6) and the antibody helping effect of GpepIP epitope is dependent on CD4 T cells (adoptive CD4+ T cell transfer experiment as well as CD4+ T cell depletion experiment) (supplemental Figure 9).

In addition, in Figure 3, we show that GpepIP specific CD4+ T cells recognize glycopeptide and/or potentially glycan epitopes. Glycopeptide and/or glycan epitopes will also be generated after Env trimer BG505 processing in APCs. So, GpepIP specific CD4+ T cells cross-react with shared glycopeptide and/or glycan epitopes from BG505 antigen. Therefore, we stated that the GpepIP-response elicited in the adjuvant group is potentially due to the BG505 Env boost in Figure 6b. This can be further supported by the evidence that pepIP, lack of shared epitopes with BG505, did not stimulate CD4+ T

cell response in the adjuvant group after BG505 Env boost. To avoid ambiguity, we have now added: “The stimulation of T cells by GpepIP in the adjuvant-primed group is likely due to the response to the shared glycopeptide/glycan epitopes from Env trimer after BG505 booster immunization.”

Taken together, we strongly believe that the enhanced responses in Fig 6d–g are due to glycopeptide specific activation of T cells rather than non-specificity.

Comment 11:

Figure 6C (and Figure 7B). Is the ELISA against the BG505 Env trimer or BG505 peptide? The title of the graph just indicates the strain name of the virus.

Response:

We have now specified as “BG505 Env IgG” in the figures.

Comment 12:

Figure 6C. Again I am surprised that there are Ab-responses against the BG505 Env trimer(?) before the trimer is given (day 28). I think this should be discussed more, especially since a peptide from a different strain (clade B) has been used as prime.

Response:

This is a very insightful point. We have added sentences in the results as “Of note, GpepIP priming yielded antibody responses against the BG505 Env trimer even before the trimer was given (day 28); this is most likely due to shared glycopeptide/glycan antibody epitopes between GpepIP and Env trimer”. We have also verified these results as part of another ongoing study that sera from GpepIP immunization can recognize BG505 trimer albeit with relatively low IgG titers. In addition, the sera from GpepIP immunization have much higher IgG titer against the same clade and cross-clade Env trimers than pepIP immunized sera.

Comment 13:

In Figure 6D it is unclear what is being measured. In the results section it is noted that this is Env trimer-specific B cell responses, but in the figure it says GC B cells - total GC B cells? It is likely that the peptide has activated B cells specific only to the peptide and not to the Env trimer and/or there can be un-specific responses going on at the same time so the B cells should be stained by flow cytometry using a biotinylated Env trimer in order to confirm they are trimer Env-specific.

Response:

We thank the reviewer for this comment. In Figure 6d, we analyzed the GL7+Fas+ (GC) B cells within total B cells. We have changed the figure 6 legend as “the percentage of GL7+Fas+ B cells among B220+ B cells” and changed the y-axis of Fig 6d as “% GL7+Fas+ of total B cells”.

Comment 14:

The last sentence in the discussion claims that GpepIP induces T cells that activate B cells with protective antibodies. I don't think the word protective is suitable since the MN.3 virus is the lowest bar of neutralization (T cell-line-adapted virus, Clade 1A) and does not reflect natural neutralization of primary viruses.

Response:

We have changed "protective" to "functional".

Comment 15:

"HIV" should be changed to "HIV-1" throughout the paper.

Response:

We have changed all "HIV" into "HIV-1" throughout the paper.

Comment 16:

In some figure legends the result of the experiment is included and should be removed.

Response:

We have removed all results from figure legends.

Comment 17:

How many times is each experiment repeated? This information should be added to figure legends.

Response:

We have added this information to every figure legend to which it applies.

Comment 18:

The colors for pepIP and GpepIP are red and blue respectively in figure 6b and d but it the opposite in figure 6e-g. It would be more comprehensive if they are the same.

Response:

We have changed the colors in figure 6 to be more consistent.

Comment 19:

The titles of the y-axes should be looked over. Fig 6d indicates for example "% of GL7+Fas+ cells" and should probably be "% GL7+Fas+ of total B cells" or "% Env-specific B cells of GC B cells".

Response:

We have changed the y-axis of Fig 6d as "% GL7+Fas+ of total B cells".

REVIEWERS' COMMENTS:

Reviewer #1 (Remarks to the Author):

In the responses and the additional data included in the manuscript, the authors have satisfactorily addressed all of my comments. The data and text now give solid support for the major claims and title of the manuscript.

Reviewer #2 (Remarks to the Author):

I find the changes and additions the authors have made to the manuscript after revision have improved the manuscript substantially from its original form. The strength of this manuscript is the finding that a glycosylated T cell epitope is able to activate CD4+ T helper cells in a glycan-dependent manner that can in turn provide help for B cells to get activated. Whether the immunogens are driving the specificity of the B cell response that is critical to generate a vaccine against HIV-1 (ie broadly neutralizing antibodies), is however yet to be determined.

Response to reviewers' comments:

Both reviewers found our revision satisfactory and did not raise any additional comment to be addressed in the manuscript. Reviewer 2 sets the stage for future investigations in her/his comment "Whether the immunogens are driving the specificity of the B cell response that is critical to generate a vaccine against HIV-1 (ie broadly neutralizing antibodies), is however yet to be determined." as is already discussed in the manuscript in detail as well. We thank the reviewers for thoroughly reading our manuscript and providing insightful comments that have contributed to this publication.

REVIEWERS' COMMENTS:

Reviewer #1 (Remarks to the Author):

In the responses and the additional data included in the manuscript, the authors have satisfactorily addressed all of my comments. The data and text now give solid support for the major claims and title of the manuscript.

Reviewer #2 (Remarks to the Author):

I find the changes and additions the authors have made to the manuscript after revision have improved the manuscript substantially from its original form. The strength of this manuscript is the finding that a glycosylated T cell epitope is able to activate CD4+ T helper cells in a glycan-dependent manner that can in turn provide help for B cells to get activated. Whether the immunogens are driving the specificity of the B cell response that is critical to generate a vaccine against HIV-1 (ie broadly neutralizing antibodies), is however yet to be determined.